# Molecularly specific detection towards trace nitrogen dioxide by utilizing Schottky-junction-based Gas Sensor

Shipu Xu [1,2,9] ✉, Xuehan Zhou [3,9], Shidang Xu[4,9], Yan Zhang[5], Yiwen Shi[5], Xuzhong Cong [3], Qijia Xu[3], Ye Tian [6], Ying Jiang [6], Hanjie Guo [1], Jinkui Zhao[1,7], Fengqiang Sun[5,8] ✉ & Hailin Peng [3] ✉

Trace NO$_2$ detection is essential for the production and life, where the sensing strategy is appropriate for rapid detection but lacks molecular specificity. This investigation proposes a sensing mechanism dominated by surface-scattering to achieve the molecularly-specific detection. Two-dimensional Bi$_2$O$_2$Se is firstly fabricated into a Schottky-junction-based gas-sensor. Applied with an alternating excitation, the sensor simultaneously outputs multiple response signals (i.e., resistance, reactance, and the impedance angle). Their response times are shorter than 200 s at room temperature. In NO$_2$ sensing, these responses present the detection limit in ppt range and the sensitivity is up to 16.8 %·ppb$^{-1}$. This NO$_2$ sensitivity presents orders of magnitude higher than those of the common gases within the exhaled breath. The impedance angle is involved in the principle component analysis together with the other two sensing signals. Twelve kinds of typical gases containing NO$_2$ are acquired with molecular characteristics. The change in dipole moment of the target molecule adsorbed is demonstrated to correlate with the impedance angle via surface scattering. The proposed mechanism is confirmed to output ultra-sensitive sensing responses with the molecular characteristic.

Trace NO$_2$ is featured with high chemical activity and its concertation variation is usually correlated with the issues among the production and life[1–4]. Nowadays the trace NO$_2$ detection mainly relies on gas chromatography-mass spectrometer technology, adsorption photometry, and gas sensing[5–12]. The former two are of ppt (volume concentration: $1 \times 10^{-12}$) detection limit but still show two issues: (i) the related device cannot be accessible for real-time monitoring and portability; (ii) the error originated from gas-concentration characterization is nontrivial due to the long-time gas collection[5,6]. In comparison, the gas sensor is integratable and featured with rapid response. Now its sensitivity is well developed and appropriate for trace gas detection. The further development of gas sensors is limited by lack of highly specific sensing mechanism[7–15]. Molecular character shall be outputted if certain kinds of the circumambient gases induced the sensing response similar to that of NO$_2$.

To achieve molecular specificity of gas sensing, the sensing mechanism such as the oxygen ionic model is developed. In this model, the oxygen molecules adsorbed can trap the electrons from the

[1]Songshan Lake Materials Laboratory, Dongguan, PR China. [2]School of Microelectronics Science and Technology, Sun Yat-sen University, Zhuhai, PR China. [3]Center for Nanochemistry, Beijing Science and Engineering Center for Nanocarbons, Beijing National Laboratory for Molecular Sciences, College of Chemistry and Molecular Engineering, Peking University, Beijing, PR China. [4]School of Biomedical Sciences and Engineering, South China University of Technology, Guangzhou, PR China. [5]School of Chemistry, South China Normal University, Guangzhou, PR China. [6]International Center for Quantum Materials, School of Physics, Peking University, Beijing, PR China. [7]The Institute of Physics, Chinese Academy of Sciences, Beijing, PR China. [8]Key Laboratory of Theoretical Chemistry of Environment, Ministry of Education, South China Normal University, Guangzhou, PR China. [9]These authors contributed equally: Shipu Xu, Xuehan Zhou, Shidang Xu. ✉e-mail: xushp7@mail.sysu.edu.cn; fqsun@scnu.edu.cn; hlpeng@pku.edu.cn

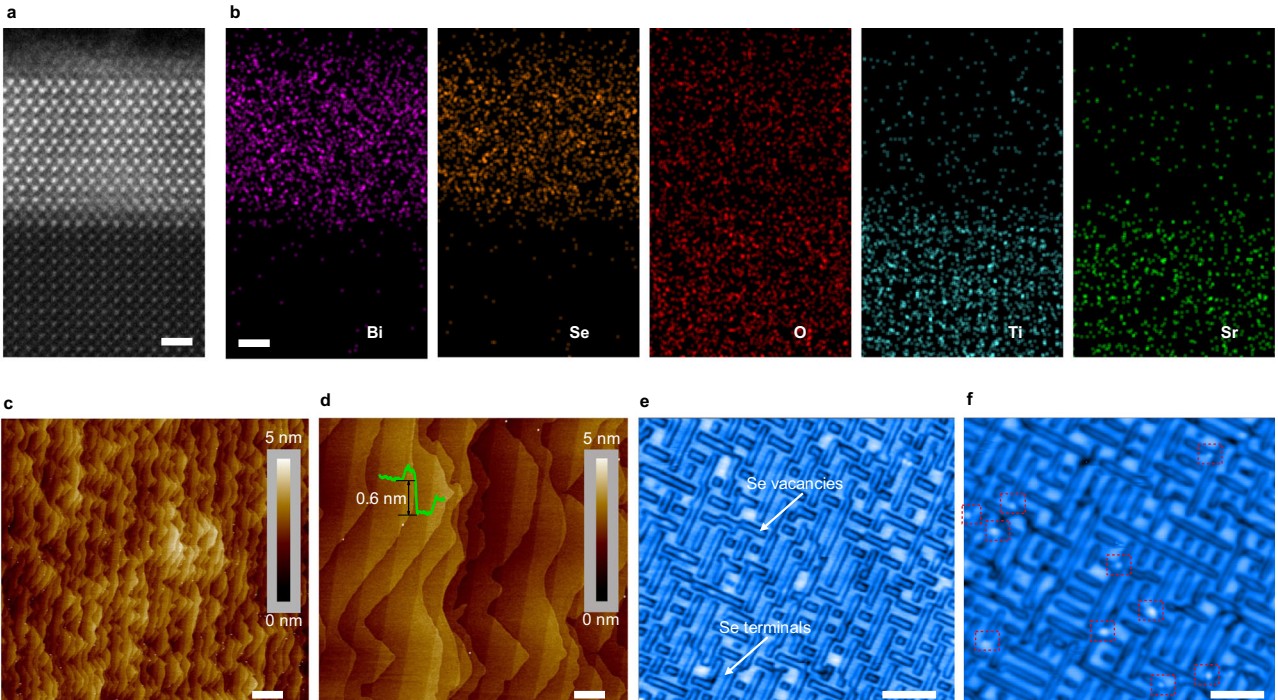

**Fig. 1 | Characterizations of Bi₂O₂Se and its adsorption active sites. a** Cross-sectional HAADF-STEM image showing the Bi₂O₂Se grown on the SrTiO₃ (001) substrate (scale bar: 1 nm). **b** The element distributions for the HAADF-STEM image (scale bar: 1 nm). The AFM images of Bi₂O₂Se surface (**c**; scale bar: 2 μm) and its zoom-in (**d**; scale bar: 400 nm). The profile in (**d**) shows 0.6-nm step-height of the Bi₂O₂Se terraces. The STM images presenting the Bi₂O₂Se surface without (**e**; scale bar: 5 nm) / with adsorbates (**f**; scale bar: 5 nm). The adsorbed O₂ molecules are indicated with red boxes.

ionic oxide to form oxygen ions[16–21]. These oxygen ions are of reducing feature and feasible to react with the oxidizing gas of NO₂, resulting in the releasing of the trapped electron back to the ionic oxide. This carrier-density variation manifests as the sensing response in the oxygen ionic model. Since this indirect surficial doping can vary the resistive signal, the NO₂ concentration can be estimated based on the resistance change. Note that the surficial doping for the sensing process manifests by the electron exchanging between the adsorbate and sensing block, rather than the internal penetration of the gas molecules into the channel. Considering that the surficial oxygen ions are highly active, certain group of the target gases may induce the similar sensing response based on the indirect surficial doping. To improve the specificity, S.-C. Chang proposes the chemical adsorption/desorption model. In this case, the direct doping occurs between the target gas and the sensing block since the adsorbed oxygen is comparatively inert at its dimer state[21,22]. Due to the energy difference between molecular hybrid orbital energy level of the gaseous molecular and affinity of the sensing block, the p/n-type doping will occur. However, the accompanying chemical reaction on the block surface induces a long-time and complex sensing response[23]. In 2015, J.H. Chen et al. fabricate the two-dimensional material (i.e., phosphorene) into a gas sensor and propose the model of surface charge transferring[13]. By the Van der Waals surface of the two-dimensional material, the sensing response is originated from the charge transferring with the negligible chemical-reaction on the sample surface. This charge transferring model is applied for the gas sensors consisted of low dimensional materials[24]. The related sensor is reported to show the ppt detection limit toward NO₂ detection[14]. Importantly, these three sensing mechanisms are based on the surface doping, and the resistive response is outputted. The resistive-type sensor cannot reveal the differential doping of each gas to export the molecular characteristic. The reason for this fact is that, even at room temperature, the heat fluctuation usually broadens the conduction band minimum of the sensing block.

To acquire molecular characteristic of trace gases, this investigation proposes surface-scattering-dominated sensing mechanism. To demonstrate its character of molecular specificity, two-dimensional Bi₂O₂Se is fabricated into a Schottky-junction-based gas-sensor and applied with alternating excitation to output multiple kinds of sensing signals simultaneously. These response signals involve resistance, reactance, and the impedance angle. Then we perform the linear fitting between the responses and gas concentration. The fitted parameters are utilized for the principle component analysis. The purpose is to reveal the correlation between the change in dipole moment of the target molecule adsorbed and the impedance angle via surface scattering. It is found that molecular characteristics are correlated with the dipole-moment variation and obtained for more than 12 kinds of target gases (containing the NO₂ mixtures). In the trace NO₂ detection, ppt-level detection limit is achieved at room temperature, and the response time is shorter than 200 s. It is confirmed that the proposed mechanism is featured with the molecular specificity and accessible for detecting trace NO₂.

## Results

### Two-dimensional Bi₂O₂Se with atomically-flat surface

Considering that Bi₂O₂Se emerges as a promising material for gas sensing[25,26], this investigation selects the Bi₂O₂Se as the sensing block. By molecular beam epitaxy, this research is to synthesize single crystalline of Bi₂O₂Se with atomic roughness on the SrTiO₃ (001) substrate. The X-ray diffraction pattern shows the lattice planes of Bi₂O₂Se on the SrTiO₃, containing (002), (004), (006), (008), and (0010) (Supplementary information S1). From the out-of-plane, electron back scatter diffraction characterization confirms that the Bi₂O₂Se is epitaxially grown along (001) crystal axis (Supplementary information S1). Figure 1a shows the high-angle annular dark-field scanning transition electron microscopy (HAADF-STEM) image of the sample in the cross-sectional view. Supporting by the characterization of element distribution, the Bi₂O₂Se is shown with

~5-nm thickness, and its growth interface is atomically sharp (Fig. 1b).

The atomic force microscopy (AFM) images in Fig. 1c, d show the $Bi_2O_2Se$ terraces with step-height of one unit cell (0.6 nm), indicating the atomic roughness of the sample. By this atomic roughness of the surface, $Bi_2O_2Se$ is bonded with the terminal of low work function (i.e., Pd; Supplementary information S2) to be characterized in terms of surface state. Supplementary information S3 shows that $Bi_2O_2Se$-Pd contact is performed with a test about current-voltage (I-V) relations at different temperatures (77–377 K). According to the thermionic emission model, the fitted barrier weight is 0.085 eV for the trivial surface state of $Bi_2O_2Se$ along the contact interface.

The scanning tunneling microscopy (STM) is conducted on the $Bi_2O_2Se$ before and after the introduction of the target gas ($O_2$; $O_2$ pressure: $1 \times 10^{-7}$ mbar) to reveal the adsorption active site. In ultrahigh vacuum, the $Bi_2O_2Se$ surface shows two kinds of terminals, i.e., the Se terminal and its vacancy (Fig. 1e). Se vacancy is active to absorb the introduced gaseous molecule rather than the Se terminal (Fig. 1f).

## Fabrication of Schottky-junction-based gas sensor

The work function of Au (5.3 eV) is higher than the Fermi level of the $Bi_2O_2Se$ (~5.2 eV, see Supplementary information S2). The $Bi_2O_2Se$ is bonded to Au electrode (Fig. 2a) to form with the Schottky junction under the forward bias, where the net electron is injected into the Au and obeys the thermionic emission theory[27]. The thicknesses of the Au electrode and the $Bi_2O_2Se$ are 50 nm and 7 nm, respectively (Fig. 2b and Supplementary Fig. 4a). Kelvin probe force microscopy (KPFM) characterizes that the surface potential between the $Bi_2O_2Se$-Au contact is 50 mV (Fig. 2c). The I-V characterizations are performed on the $Bi_2O_2Se$-Au contact at different temperatures (T) ranging from 280 K to 310 K, and these measurements are respectively conducted in vacuum and in 700 ppt $NO_2$. The I-V curves in vacuum/$NO_2$ environment present none-linear correlation for the existence of the Schottky junction, and the output current is positively correlated with the applied voltage (Fig. 2d–i). As the temperature increases from 280 K to 320 K, the sample resistance increases. It is indicated that the highly-doped characteristic of the $Bi_2O_2Se$ suppresses the effect of the Schottky barrier height (SBH)[28]. Note that the resistance in $NO_2$ is higher than that in vacuum, indicating the p-type doping/surface scattering originated from the adsorbed $NO_2$[29].

The variation of SBH is estimated by fitting ln $(I/T^2)$ and $T^{-1}$, according to the thermionic emission theory. In details, the thermionic emission theory describes the correlation between the current I and SBH ($\Delta E$) as follows[27]:

$$I = SA^*T^2 \exp((-\Delta E)/(k_B T))(\exp((qV_{ds})/(k_B T))-1) \quad (1)$$

where S, $A^*$, q, and $k_B$ are respectively contact area, Richard coefficient, charge quantity, and Boltzmann constant. Note that when the applied voltage $V_{ds}$ is large enough and the Eq. (1) can be simplified as Eq. (2) below:

$$I = SA^*T^2 \exp((qV_{ds} - \Delta E)/(k_B T)) \quad (2)$$

where the slope extracted by the linearly fitting between ln $(I/T^2)$ and $T^{-1}$ is served as the value of $(qV_{ds} - \Delta E)/k_B$, and the SBH value $\Delta E$ is obtained with the certain applied voltage $V_{ds}$. In $NO_2$, the $\Delta E$ ($NO_2$) is then estimated to be 2.795 eV ($V_{ds}$: 3 V) which is lower than that in vacuum ($\Delta E$ (Vac.): 2.846 eV). It is indicated that the adsorbed $NO_2$ decreases the carrier density of the $Bi_2O_2Se$ ($n_0$) and the SBH reduces by 0.051 eV.

To reveal this carrier-density-dominated barrier variation, the $Bi_2O_2Se$ is fabricated into a Hall device for temperature-dependent resistance measurement (Supplementary information S4). The temperature range is 2–380 K, and the test is conducted in Ar. Figure 2j and Supplementary Fig. 4c show that the resistance (R) is positively

correlated to the temperature (T). This metal-like R-T behavior indicates the highly-doped feature of the $Bi_2O_2Se$, where this highly-doped character is reported to be originated from the self-modulation doping effect[28]. The highly-doped feature of the $Bi_2O_2Se$ is further supported by the negative relation between the mobility of $Bi_2O_2Se$ and temperature (Fig. 2k) since the temperature rise enhances the carrier scattering effect.

Considering that the carrier density variation of $Bi_2O_2Se$ is originated from the p-typed doping of the $NO_2$ adsorption, its thermodynamic feasibility can be justified by analyzing the ionization energy of $Bi_2O_2Se$ ($\Delta E_D$). Herein, the ionization energy of $Bi_2O_2Se$ is extracted by linearly fitting the value of ln($n_0$) and $1/T$ when the operating temperature is low, obeying the formula below:

$$n_0 = (N_D N_c/2)^{1/2} \exp((-\Delta E_D)/(2k_B T)) \quad (3)$$

where $N_D$ and $N_C$ are donor concentration and effective state density of conduction band, respectively. Then the temperature-dependent carrier-density variation is investigated for the $Bi_2O_2Se$ in vacuum as the temperature decreases from 64 K to 50 K. Figure 2l presents the linear fitting based on the values of ln($n_0$) and $1/T$, then the slope is extracted as $-E_D/2k$ or $-E_D/k$. The ionization energy of 0.025 eV is acquired for the highly-doped character of the $Bi_2O_2Se$. By this low ionization energy and the highly-doped feature aforementioned, the p-type doping of the adsorbates is highly feasible to occur on the $Bi_2O_2Se$ at room temperature.

## Molecularly specific detection towards trace gases

In the trace gas detection, the Schottky-junction-based gas sensor is equivalent to a circuit formed by a resistance ($R_{sens.}$) parallelly connected with a capacitance ($C_{sens.}$; Fig. 3a). Then this sensor is serially connected with a resistance ($R_s$: 5 MΩ). The excitation is an alternating voltage of 0.5 V with frequency of 400 Hz. To characterize the barrier variation for the gas introduction, the frequency-dependent excitation is applied on the sensor with/without $NO_2$. The reactance signal is measured and shown in Fig. 3b. The extreme point of the frequency-dependent reactance is shifted from the high frequency to the low for the $NO_2$ introduction. This shifting indicates an extended relaxation time for the $NO_2$ adsorption, where the relaxation time originates from the capacitance feature of the Schottky junction. In model of double-well energy[30], the relaxation time is positively correlated to the barrier height. It is indicated that the Schottky barrier height increases for the $NO_2$ introduction. Hence, by the alternating excitation, the Schottky junction can be also modulated by surface adsorption.

Applied with the alternating excitation, the Schottky-junction-based gas sensor simultaneously outputs three kinds of sensing signals. These signals involve resistance, reactance, and impedance angle (R, X, and Θ). These responses are sensitive to the trace gas (e.g., methanol, methanal, and acetone at ppm-ppb range) at room temperature (Fig. 3c, d, and Supplementary information S5). And the signal variations are positively correlated to the gas concentration. Related to the methanal and acetone, the absolute values of resistance and reactance increase for the p-type doping of adsorbates (see Supplementary Table 1). For the methanol, its surface scattering effect suppresses the n-type doping, by which the sensor resistance increases. Regarding the impedance angle, it is dominated by the surface scattering. The absolute value of the impedance angle increases for more adsorbates (e.g., methanol, methanal, and acetone). More details about the correlation between the surface scattering and impedance angle are illustrated in Supplementary information S6. Figure 3c, d present that, both response times are shorter than 200 s and the recovery times are around 100 s. According to Einstein equation, the slopes of the sensing responses are the self-diffusion coefficients. In the case of acetone detection (see Supplementary information S5), the slopes of the reactance and impedance are linearly and positively

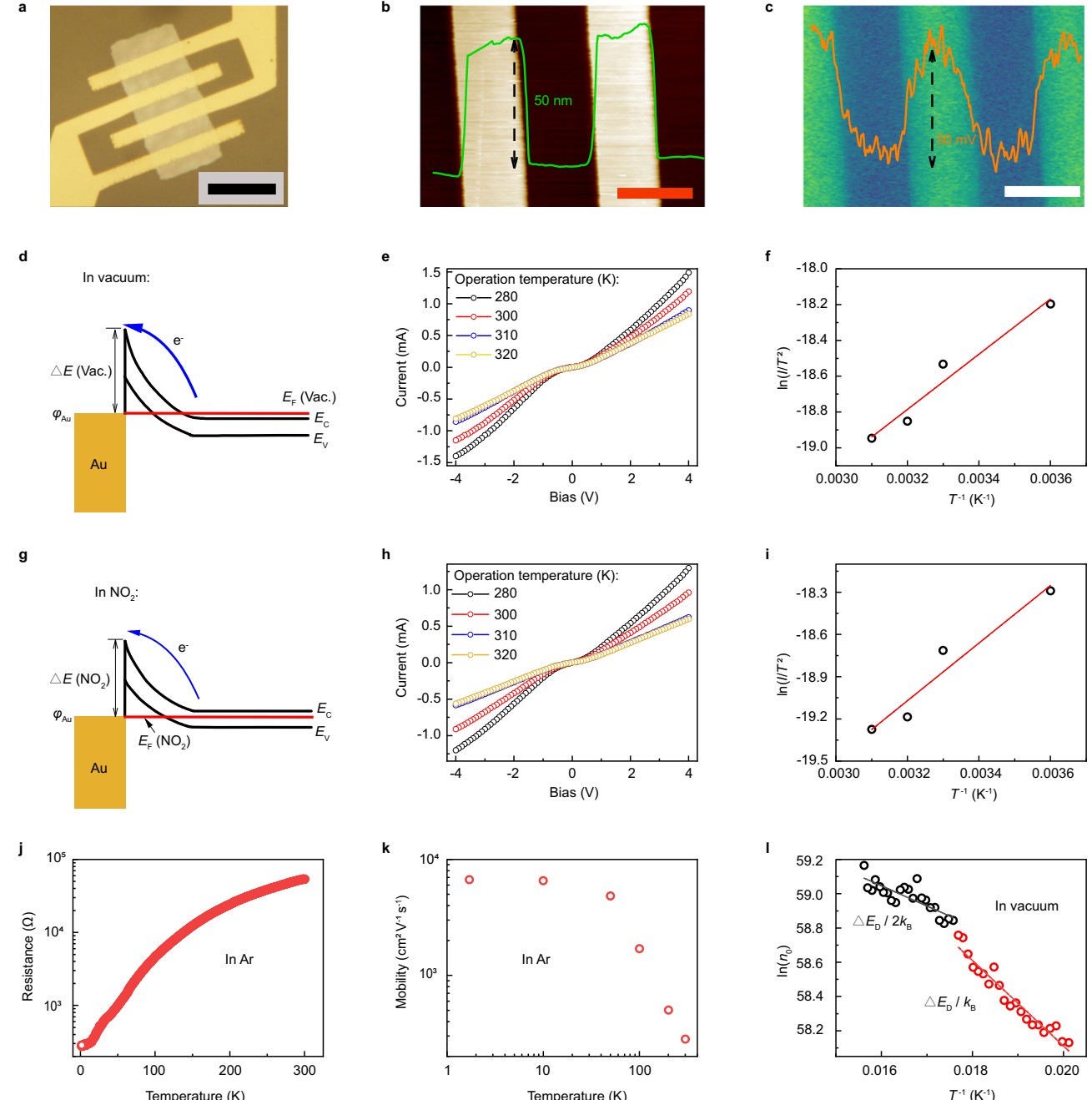

**Fig. 2 | Adsorbate-induced variation of the Schottky-junction barrier. a** The photograph of the Bi$_2$O$_2$Se gas sensor obtained by utilizing optical microscopy (scale bar: 12 μm). **b** AFM image showing topology of the Bi$_2$O$_2$Se-Au contact (scale bar: 3 μm). **c** The KPFM pattern of Bi$_2$O$_2$Se-Au contact (scale bar: 3 μm). **d** The energy band profile of the forward-biased contact in vacuum (vac.) ($\varphi_{Au}$: work function of Au; $E_F$: Fermi level of Bi$_2$O$_2$Se; $E_c$: conduction band minimum; $E_v$: valence band maximum). **e**, **f** The temperature-dependent *I-V* plot of the device in vacuum and its linear fitting (**f**, ln($I/T^2$) and $T^{-1}$). **g** The energy band profile of the forward-biased contact in NO$_2$ (700 ppt). **h** The *I-V* relation as the function of temperature in NO$_2$ (700 ppt). **i** The linear fitting between ln($I/T^2$) and $T^{-1}$ for *I-V* curve in (**h**). The temperature-dependent resistance (**j**) and mobility (**k**) in Ar environment. **l** The linear fitting for extraction of ionization energy (the measurement conducted in vacuum).

correlated with the acetone-concentration rather than that of the resistive. It is indicated that, the resistance response is dominated by multiple elements such as carrier density and mobility.

To demonstrate the molecular specificity of the impedance angle, 12 kinds of gases common in human exhaled breath together with NO$_2$ mixtures are served as target gases. These target gases include NO$_2$, O$_2$, NH$_3$, and the volatile organic compounds such as acetone. Figure 4a shows the responses to isopropanol, acetone, and NO$_2$. Their response time is less than 200 s at room temperature. Towards the 400 ppt NO$_2$,

the response is 3.5/3/6.7 % for the resistive/reactance/impedance signal. Considering that the noise is lower than 0.4 % at the frequency range higher than 0.02 Hz (Supplementary information S7), this signal-to-noise ratio indicates that the NO$_2$ sensing is featured with the detection limit in ppt range. The sensitivity is estimated to 8.8/7.5/16.8 %·ppb$^{-1}$ for the resistive/reactance/impedance. The NO$_2$ sensitivity shows orders of magnitude higher than those of the other gases. The ultrahigh sensitivity of the resistive and reactance is attributed to the fact that, p-type doping of NO$_2$ is distinct as compared against those of the others (see

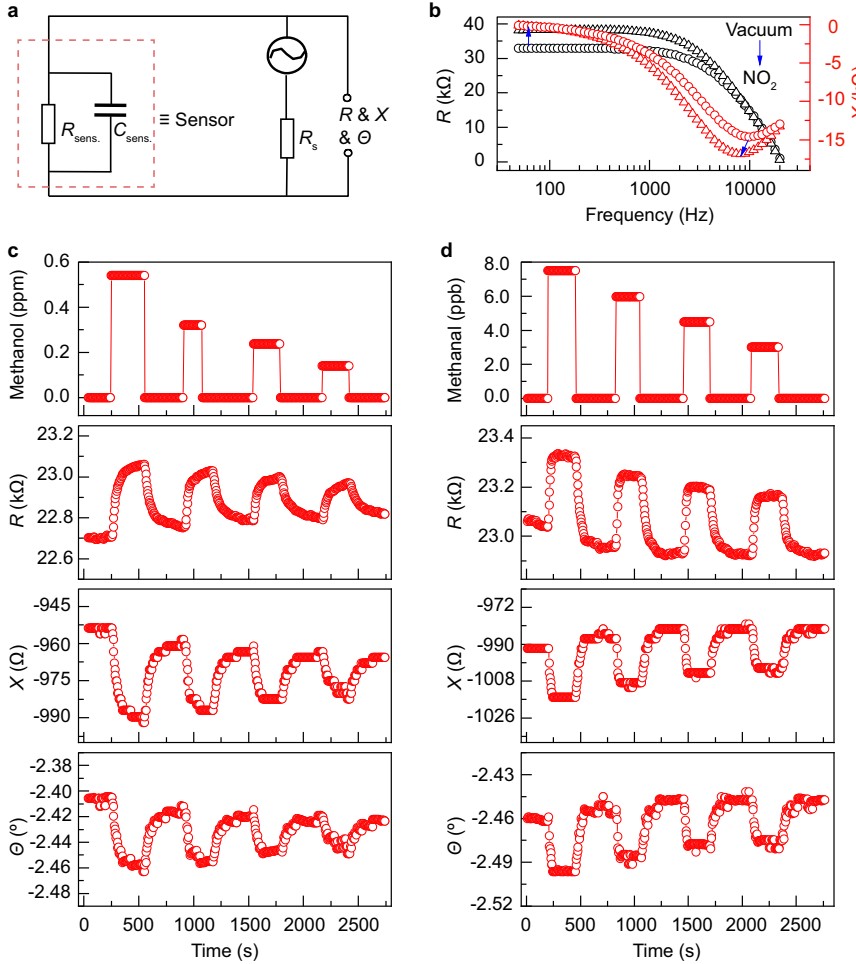

**Fig. 3 | The simultaneous outputting of multiple sensing signals. a** The schematic showing trace gas detection by utilization of the $Bi_2O_2Se$ sensor. This $Bi_2O_2Se$ sensor is featured with the Schottky junction and equivalent to a circuit consisted of a resistance ($R_{sens.}$) and a capacitance ($C_{sens.}$). **b** The frequency-dependent $R$ and $X$ in vacuum/$NO_2$. The blue arrows indicate the environment changing from the vacuum to the $NO_2$. The multiple sensing signals ($R$, $X$, and $\Theta$) towards the methanol (**c**)/methanal (**d**) exposure.

Supplementary information S6). Note that this $NO_2$ doping manifests as the charge transferring from the $Bi_2O_2Se$ to the $NO_2$ adsorbed. The $NO_2$ is therefore served as the adsorbate, rather than the dopant for inducing the reconstruction of the $Bi_2O_2Se$ (Supplementary information S8). The ppt-response of impedance angle is attributed to the apparent change in dipole moment of the $NO_2$ adsorbed, which is positively related to the surface scattering (Supplementary information S6).

The molecular specificity of the impedance angle is further demonstrated when the target gas $NO_2$ is mixed with the other gas. Firstly, the trace $NO_2$ (430 ppt) is mixed with the low concentration $O_2$ ($C(NO_2)/C(O_2) = 1 \times 10^{-4}$) and served as target gas in Fig. 4b. Compared to the responses to the $NO_2$ target gas, the resistive sensitivity to the mixture almost remains (10.5 %·ppb$^{-1}$), while the reactance and the impedance-angle sensitivity are specific to the mixed gas (the reactance: 23.3 %·ppb$^{-1}$; and the impedance: 12.1 %·ppb$^{-1}$). To further study on the molecular specificity, the concentration of the mixed (e.g., methanol or methanal) is increased up to 33% and 50 %. Figure 4b presents that, in comparison with response to the $NO_2$, the sensitivities of the multiple signals reduce for the gas-mixture detection, indicating the existence of the mixed gases. In the case of the $NO_2$ mixed with 50 % of methanal, the sensitivities are respectively decreased by 75 %, 30 %, and 80 % for those of the resistance, the reactance, and the impendence angle. In contrast to the sensing signals of the resistance and the reactance, the impedance angle is more sensitive to each gas in the mixture and featured with molecular specificity.

To output molecular characteristic, this research conducts the principle component analysis based on the multiple sensing signals (Fig. 5a, b). In this way, the sensing signals can be transferred into a new coordinate axis, where the component projected on each axis can be displayed with the maximum variation. It is therefore more distinct for the molecular characteristic presentation since the sensing signals for each molecule can be sufficiently dispersed. It means that the molecular characteristic can be differential by the well-dispersed component projected on each axis and applied for molecularly specific detection. In the principle component analysis, the responses of the multiple signals are firstly linearly fitted with the corresponding gas-concentration. The fitted parameters involving the slope and the intercept are obtained for different target gases (see Supplementary Table 2). These parameters are analyzed by the principle component analysis. Figure 5a shows the three-dimensional component plot based on the multiple sensing signals. The projections on the principle component (PC) 1–3 are differential for each gas. These none-overlayed projections can be served as the molecule characteristics. In contrast, the projections are highly overlayed for the principle component analysis based on the fitted data of the resistance (i.e., slope and intercept originated from the linear fitting), where only is two-dimensional plot presented in Fig. 5b. The reason for this fact is that, in the principle component analysis, the variance (a) originated from the multiple sensing signals is a$^6$ (a > 1), which is 4 orders of magnitude larger than that of the traditional resistance. Thus the

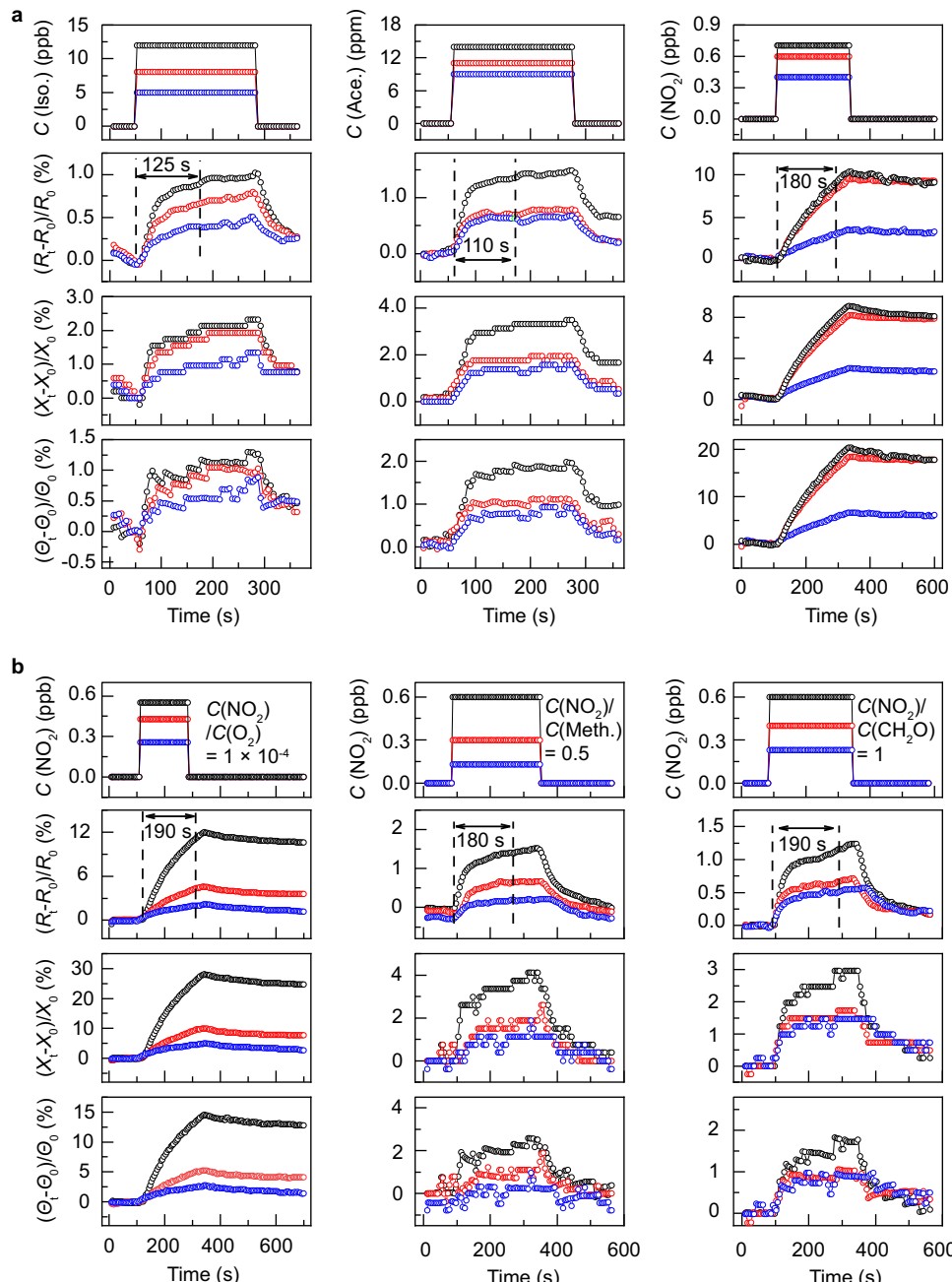

**Fig. 4 | Sensing responses to different gases.** The responses (estimated by signals of $R$, $X$, and $\Theta$) to different gases (**a**; isopropanol (Iso.), acetone (Ace.), $NO_2$) and $NO_2$ mixture (**b**; $NO_2/O_2$, $NO_2/CH_3OH$ (Meth.), $NO_2/CH_2O$). The concentrations of the target gas are arranged from the high to the low and sequentially notated in black, red, and blue. The sensing responses to the target gases with different concentrations are marked with the corresponding colors.

projections on the PC 1–3 can be differential to show the molecular characteristics. In this mean, this study further applies the principle component analysis to output the molecular characteristic of $NO_2$ mixture. Figure 5a presents that, each mixture is featured with specific molecular characteristic and adjacent to that of $NO_2$, by which the contained $NO_2$ can be detected.

## Discussion

By exploring multiple sensing signals originated from the Schottky-junction-based gas-sensor, this investigation proposes the surface-scattering-dominated sensing mechanism and proves it with feature of molecular specificity. The molecular selectivity is proved to be correlated with the change in dipole moment of target molecules adsorbed and manifests as the impedance angle. The molecular characteristics

for pure gases and the mixture are then outputted based on the principle component analysis. 12 kinds of target gases common in human exhaled breath are characterized with molecule features. The $NO_2$ detection is featured with detection limit in ppt range. And the $NO_2$ sensitivity is reached up to 16.8 %·ppb$^{-1}$. This $NO_2$ sensitivity shows orders of magnitude higher than those of the other common gases. The proposed sensing paradigm is therefore highly accessible for detecting the exhaled disease markers and especially for the $NO_2$ sensing.

## Methods

### Preparation of Bi$_2$O$_2$Se film

A home-made oxide molecular beam epitaxy (OMBE) system is applied for the $Bi_2O_2Se$ films growth. Bismuth (99.999%) and selenium (99.999%) are sources. $SrTiO_3$ (001) single crystals (Shinkosha) are the

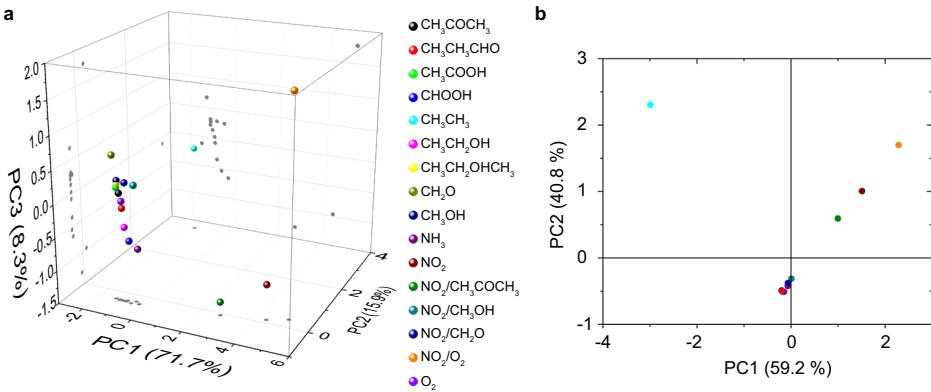

**Fig. 5 | Analysis of molecular characteristics.** The principle component analysis based on the multiple signals (**a**; containing those of $R$, $X$, and $\Theta$) and the $R$ (**b**).

growth substrates. Firstly, to obtain single TiO$_2$-terminated surface, an annealing is conducted on the SrTiO$_3$ substrates in ultrahigh vacuum at 950 °C for 45 min. To eliminate the formed oxygen vacancies, the substrate is annealed at 600 °C for 15 min with the oxygen pressure of $1.3 \times 10^{-4}$ mbar. Co-evaporating bismuth and selenium precursors under the oxygen pressure of $1.3 \times 10^{-4}$ mbar, the Bi$_2$O$_2$Se film is synthesized and the substrate temperature is kept at 380 °C. In this case, the selenium source is heated at 80 °C and the bismuth source is maintained at 610 °C with a flux of ~0.75 Å/min; the Bi$_2$O$_2$Se thickness is dependent on growth time. Finally, the sources shutters are simultaneously closed and then the sample is cooled down to the room temperature naturally.

### Characterizations of Bi$_2$O$_2$Se
To characterize Bi$_2$O$_2$Se, this investigation conducts optical microscopy (Olympus DX51 microscope), AFM (Bruker Dimension Icon), KPFM (Oxford Instruments Asylum Research, Cypher S), the STM (Createc), XRD (Rigaku Dmax 2500 PC, Cu Kα radiation), X-ray photoelectron spectroscopy, STEM aberration-corrected transmission electron microscope at 300 kV (FEI Titan Cubed Themis G2).

### Preparation of Bi$_2$O$_2$Se sensor
Firstly, the research applies photolithography technique to predefine alignment markers on SrTiO$_3$ substrate (001) grown with Bi$_2$O$_2$Se. The electron-beam lithography is then utilized for the electrode writing. Au is served as the electrode material and deposited with 50-nm thickness. The Bi$_2$O$_2$Se sensor is acquired after the photoresist removal.

### Gas sensing measurement
The gas-sensing configuration contains the dynamic intake and the electrical test system. The gas intake system mainly consists of the mechanical and the molecular pump, by which vacuum degree of $1 \times 10^{-8}$ mbar can be reached for dilution of target gas down to the ppt concentration.

In this mean, the initial gas is 100 ppm ($C_0$) target gas mixed with N$_2$ (gas pressure $P_0$: 10 MPa; purchased from Air Liquide). Regarding the dilution, this initial gas is injected into the test chamber with a comparatively low gas pressure ($P$, ranging from $1 \times 10^{-4}$ Pa to $1 \times 10^4$ Pa); the diluted concentration of target gas ($C$) is estimated according to the following equation:

$$C = (P/(1\,atm))C_0\beta$$

where $\beta$ is correction coefficient ($\beta = 1 \times 10^{-3}$; Supplementary information S10) and the diluted concentration is considered to be featured with the gas pressure of 1 atm.

The electrical test system is based on a commercial locking-in (SR 830). With an alternating voltage of 0.5 V with frequency of 400 Hz, the

sensing signals contain the resistance, reactance, and impedance angle. Without the target gas, the resistance, reactance, and impedance angle signal are respectively collected as $R_0$, $X_0$, and $\Theta_0$. Introducing target gas, this investigation measures these three signals till their stable stages are reached. Then target gas is removed for the recovery of the sensor signals. The response is defined as $(X_t - X_0)/X_0$ or $(R_t - R_0)/R_0$ or $(\Theta_t - \Theta_0)/\Theta_0$. The sensitivity is estimated by $(X_t - X_0)/(X_0 \cdot C_{target})$ or $(R_t - R_0)/(R_0 \cdot C_{target})$ or $(\Theta_t - \Theta_0)/(\Theta_0 \cdot C_{target})$, where is $C_{target}$ the concentration of target gas.

### Reporting summary
Further information on research design is available in the Nature Portfolio Reporting Summary linked to this article.

## Data availability
Source data for this paper are deposited in Figshare (ref. 31).

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

## Acknowledgements

S.P.X. is grateful to supports provided by National Natural Science Foundation of China (62304145) and Guandong Basic and Applied Basic Research Foundation (grant number: 2021A1515110313 and grant number: 2024A1515010951). H.L.P. acknowledges supports from the National Key Research and Development Program of China (2022YFA1204900) and the National Natural Science Foundation of China (21920102004). F.Q.S is grateful to supports from GDUPS (2019). S.D.X thanks to supports from National Natural Science Foundation of China (52373136) and National Key Research and Development Program of China (2022YFB3804700). Y.J. acknowledges supports from The National Key R&D Program (grant number: 2021YFA1400500) and The National Natural Science Foundation of China (grant number: 11888101 and grant number: 21725302).

## Author contributions

S.P. Xu designed the research then carried out the experiments and wrote the manuscript. H. L. Peng guided the synthesis of $Bi_2O_2Se$ and its device construction into the Schottky-junction-based sensor, as well as conceived the exploration of $Bi_2O_2Se$ for gas-sensing applications. X.H. Zhou synthesized $Bi_2O_2Se$ and fabricated it into the gas sensor, together with assistance from X.Z. Cong and Q.J. Xu. S.D. Xu provided theory data. Y. Tian and Y. Jiang conducted STM works. Y. Zhang, Y.W. Shi, and F.Q. Sun supported the characterization of gas sensors. F.Q Sun assisted S.P. Xu in the manuscript writing. H.J. Guo and J.K. Zhao supported the building of the experimental set-up. All the researchers participated in discussion of this research.

## Competing interests

The authors declare no competing interests.
