## [Peer Review File · Nature Communications]

Reviewers' Comments:

Reviewer #1:

Remarks to the Author:

This paper provides an innovative approach for characterizing multiple gases simultaneously in a mixed gas environment.

To this end, the paper presents the relationship between the change in impedance, mobility, and the dipole moment that influences this mobility change. Additionally, it analyzes the change in resistance, reactance, and impedance angle in an actual mixed gas environment, adding persuasiveness to the hypothesis.

This paper certainly makes an important contribution and has academic value. However, before publication, there are a few key aspects that need to be more clearly addressed and improved. If these areas are strengthened, I expect that the paper's clarity and academic depth will be further enhanced.

Q1. Key Equations: The manuscript does not mention several critical equations, such as SBH and $E_d/2k$. I recommend including a detailed explanation of the used equations and calculation methods. Additionally, it is needed to mention V_{ds} in the context of the Arrhenius equation.

Q2. Figure 2 (e, h): Typically, in a device dominated by the thermionic emission (TE) mechanism, the device resistance should decrease as temperature increases. However, the behavior observed in Figure 2 (e, h) appears contradictory. Could you clarify the reason behind this?

Q3. Figure 2 (f, i): The method and analysis for extracting the Schottky Barrier Height (SBH) from Arrhenius plots seem incorrect. The negative slope of the plot should represent SBH, and accurate extraction of SBH requires suppression of thermionic emission effects, especially at lower temperatures. Furthermore, extracting SBH correctly is possible at the flat band V_g point. The temperature range used in the current analysis seems too narrow and inadequate to exclude TE effects.

Q4. Figure 2 (j, k, l): The gas environment is not clearly stated (presumably vacuum). Also, for Figure 2 (l), consider revising the notation to $(1/T)$.

Q5. Reference: Please provide a reference for the inverse relationship between temperature and mobility at high doping concentrations, as observed in Fig 2j.

Q6. Explanation and Clarity in Figure 5: The description in Figure 5 is unclear. A more detailed and understandable explanation of the principle component and its implications is necessary. As Nature Communications is a general science journal, the explanation should be accessible to a broader audience.

Q7. Statement in Section S6: The statement, "the adsorbate with weaker dipole moment shows shorter binding length, indicating a larger quantity of charge," seems contradictory to typical understandings of dipole moments in molecules. Typically, a weaker dipole moment implies less charge separation, which might not necessarily correlate with shorter binding lengths or larger charge quantities.

A more detailed explanation or a simplified statement such as, "The charge induced on the surface by the dipole moment acts as a scattering center, thus changes in dipole moment can lead to changes in mobility," would be more appropriate.

Q8. Inconsistency in Section S6: There is a contradiction in stating that "The charge quantity is positively related to the binding length between the target molecule and the sensing block." but concluding that a lower dipole results in a higher charge. This needs clarification or correction.

Q9. In page 4, line 117, Please revise the expression.

Figure 1a shows the high angle annular dark-field scanning transition microscopy (HAADF-STEM) image of the sample from the cross-sectional view.

Q10. In page 6, line 150, Please revise the expression.

"The I-V characterization is conducted in vacuum/700 ppt NO₂ at different temperature (T, ranging in 280-310 K)"

Q11. Bi₂O₂Se is a promising material for various gas sensing reactions, as can be seen in the paper "One-Step Synthesis of Bi₂O₂Se Microstructures for Trace Oxygen Gas Sensor Application" by Ji kwon Bae, Hyeon Ho Cho, et al. published in *Sensors & Actuators: B* (2023). indicates that Therefore, the author recommends adding an application for Bi₂O₂Se by referring to various papers

(https://www.sciencedirect.com/science/article/pii/S0925400523011139?casa_token=lxrctiByKswAAAAA:ItY_o02GI08IiJvPoDXk7m1ePai8bItU7IR2vJfIPBEpPvnwdKs-wPOM hmXcdkZ5Yjx_sJs) This study is a single Bi₂O₂Se microstructures are successfully synthesized using a zoned chemical vapor deposition (CVD) method, demonstrating their effectiveness as O₂ sensors.

Q12. One of the important properties of a gas sensor is its stability. The gas sensor should ensure stability over a long period of time even with various times of gas exposure. The authors should supplement with additional data on stability.

Q13. In Figure 2, the author set the temperature range to 2-300K to explain resistance according to temperature. However, correlations should be investigated not only at temperatures below the room temperature but also at temperatures above the room temperature.

- Recommendations for Strengthening the Manuscript:

R1. Demonstrating the change in mobility due to the dipole moment through calculations could significantly enhance the persuasiveness of the paper.

R2. Given that Nature Communications covers multidisciplinary science, a more detailed and easily understandable explanation throughout the manuscript would be beneficial.

Reviewer #2:

Remarks to the Author:

The manuscript entitled "Molecularly Specific Detection towards Trace Nitrogen Dioxide by Utilizing Schottky-Junction-based Gas Sensor" proposes a crucial issue in the current gas sensor research, i.e., the lack of highly specific sensing mechanism for the semiconductor-based chemiresistive and chemical field-effect transistor sensors, which leads to the poor selectivity of the sensor toward specific target gas. Many problems existed in the manuscript and my comments are as follows:

1. In the Introduction part, "NO₂ molecules firstly react with these oxygen ions to release the trapped electrons back to the sensing block." This description is absolutely wrong, since NO₂ is an oxidizing gas. The authors classified three sensing mechanisms in this part, i.e., "oxygen ionic model", "chemical adsorption/desorption model" and "the model of surface charge transferring". Actually, these three mechanisms are in essence the same. For practical operation of a gas sensor, interaction with oxygen cannot be excluded, so "direct doping is formed between the target gas and the sensing block" is rare. A sensing process always includes gas adsorption and charge transferring due to gas-solid interaction.

2. In Fig. 1e and f, the authors present the STM images regarding the adsorption of O₂ molecules on the Se vacancies. Firstly, the sites for Se terminals/vacancies and all the adsorbed O₂ molecules should be clearly indicated on the STM images. Secondly, does this structure character have any relationship with the proposed gas sensing mechanism (dipole moment of molecule)?

3. "Note that the resistance in NO₂ is higher than that in vacuum, indicating the p-type doping/surface scattering originated from the adsorbed NO₂." The authors describe the increased resistance in NO₂ due to "p-type doping", do they mean NO₂ molecules are doped onto the surface

of Bi₂O₂Se? If so, then the sensor will have a problem to recover, as a dopant in the material is relatively stable.

4. In Fig. 2j, "the resistance is positively correlate to the temperature. The highly-doped feature of the Bi₂O₂Se is therefore clarified", the authors are suggested to explain clearly the high doping in Bi₂O₂Se.

5. Fig. 3, "The resistance and reactance value increase for the p-type doping of adsorbates." This is not the fact, as one can see in Fig. 3c and d that reactance decreases upon exposing to methanol/methanal.

6. Figs. 3 and 4, why the resistance of Bi₂O₂Se increases upon exposing to typical reducing gases such as methanol, methanal, isopropanol and acetone. These reducing gases behaves just like oxidizing NO₂.

7. Poor description in the manuscript, e.g., "the sensitivity of the resistive", "As the concentration of the mixed increases from 33% to 50 %, the sensitivities of the multiple signals (involving R) reduce for the NO₂ detection".

8. In S6, "(1) the surface scattering usually interferes the resistive signal and degrades the sensitivity;" Take the response of Bi₂O₂Se toward NO₂ for an example, on the one hand, "the adsorbed NO₂ decreases the carrier density of the Bi₂O₂Se (n)", on the other hand, the adsorbed NO₂ molecules form as the scattering centers and reduce the carrier mobility (μ) of Bi₂O₂Se. As both n and μ decrease, a even higher resistance is obtained for Bi₂O₂Se, and a larger response to NO₂ is achieved. Therefore, in this case, the surface scattering does not interfere the resistive signal, but reinforces it.

9. The authors claimed that they achieved "Molecularly Specific Detection of NO₂" by measuring the impedance angle, which is dependent on the dipole moment of the target molecule and featured with molecular specificity. However, as seen in Fig. 4, all gases can give a signal for the parameter of "impedance angle", which can seriously interfere the detection of NO₂. The method to reach "Molecularly Specific Detection of NO₂" is by PCA, an approach based on statistical analysis. This means the method proposed in this work is not molecularly specific, but a result of statistical analysis. A typical example for molecularly specific detection is non-dispersive infrared gas sensors, which analyze the concentration of target gases based on their characteristic infrared absorption. The detection of a target gas cannot be interfered by other ones.

Reviewer #3:

Remarks to the Author:

The manuscript describes the development of a sensor device for determining ultra-low concentrations of nitrogen dioxide. The authors propose to use the impedance angle, which exhibits maximum changes when exposed to gas, as a sensor signal. This experimental fact is explained based on the model of surface scattering of charge carriers during adsorption of NO₂ molecules. In order to ensure the specificity of the gas detection, signals were processed using the of the principal component analysis. The practically important results are related to the choice of two-dimensional bismuth oxyselenide as a sensor material and the appropriate electrodes for the formation of the Schottky barrier. The introduction confirms the importance of this research. A comprehensive characterization of the synthesized material was carried out, all the conclusions were explained.

The authors should improve the article, as the manuscript contains quite a lot of grammatical errors. For example, there is a repeated error in the word "impedance". It is also necessary to correct some mistakes in the presented figures: Figure S1(c) is uninformative; Figure S2(b) does not show the peak of Se at 53 eV. Since the exhaled air always contains a large amount of water vapors, it is interesting to investigate their effect on the trace NO₂ detection characteristics.

Point-by-point response to the reviewer comments

We would like to highly appreciate the reviewer for their comments. These comments are very important for strengthening the manuscript, we have responded them point-by-point. The author reply and the corresponding updates are presented below for reading convenience.

The reviewer comment is marked in violet. Our reply and the updates are indicated in black and blue, respectively.

REVIEWER COMMENTS

Reviewer #1 (Remarks to the Author):

This paper provides an innovative approach for characterizing multiple gases simultaneously in a mixed gas environment.

To this end, the paper presents the relationship between the change in impedance, mobility, and the dipole moment that influences this mobility change. Additionally, it analyzes the change in resistance, reactance, and impedance angle in an actual mixed gas environment, adding persuasiveness to the hypothesis.

This paper certainly makes an important contribution and has academic value. However, before publication, there are a few key aspects that need to be more clearly addressed and improved. If these areas are strengthened, I expect that the paper's clarity and academic depth will be further enhanced.

Author reply: We would like to highly appreciate the reviewer for the comments. These comments have been replied point-by-point, and the manuscript has been revised accordingly.

Q1. Key Equations: The manuscript does not mention several critical equations, such as SBH and $E_d/2k$. I recommend including a detailed explanation of the used equations and calculation methods. Additionally, it is needed to mention V_{ds} in the context of the Arrhenius equation.

Author reply: Many thanks to the reviewer for this suggestion, and we have included the detailed explanation in the revised manuscript. Please read the related context below for convenience, where the V_{ds} is mentioned:

In section “**Fabrication of Schottky-Junction-based Gas Sensor**” of the manuscript: “The variation of SBH is estimated by fitting $\ln(I/T^2)$ and T^{-1} , according to the thermionic emission theory. In details, the thermionic emission theory describes the correlation between the current I and SBH (ΔE) as follows:

$$I = SA^*T^2 \exp\left(\frac{-\Delta E}{k_B T}\right) \left[\exp\left(\frac{qV_{ds}}{k_B T}\right) - 1\right] \quad (1)$$

where S , A^* , q , and k_B are respectively contact area, Richard coefficient, charge quantity, and Boltzmann constant. Note that when the applied voltage V_{ds} is large enough and the equation (1) can be simplified as equation (2) below:

$$I = SA^*T^2 \exp\left(\frac{qV_{ds} - \Delta E}{k_B T}\right) \quad (2)$$

where the slope extracted by the linearly fitting between $\ln(I/T^2)$ and T^{-1} is served as the value of $(qV_{ds} - \Delta E)/k_B$, and the SBH value ΔE is obtained with the certain applied voltage V_{ds} . In NO_2 , the ΔE (NO_2) is then estimated to be 2.795 eV (V_{ds} : 3 V) which is lower than that in vacuum (ΔE (Vac.): 2.846 eV). It is indicated that the adsorbed NO_2 decreases the carrier density of the $\text{Bi}_2\text{O}_2\text{Se}$ (n_0) and lowers $\text{Bi}_2\text{O}_2\text{Se}$ Fermi level, where the SBH reduces by 0.051 eV.

To reveal this carrier-density-dominated barrier variation, the $\text{Bi}_2\text{O}_2\text{Se}$ is fabricated into a Hall device for temperature-dependent resistance measurement (supplementary information S4). The temperature range is 2-380 K, and the test is conducted in Ar. Fig. 2j and Fig. S4c show that the resistance (R) is positively correlated to the temperature (T). This metal-like R - T behavior indicates the highly-doped feature of the $\text{Bi}_2\text{O}_2\text{Se}$, where this highly-doped character is reported to be originated from the self-modulation doping effect²⁷. The highly-doped feature of the $\text{Bi}_2\text{O}_2\text{Se}$ is further supported by the negative relation between the mobility of $\text{Bi}_2\text{O}_2\text{Se}$ and temperature (Fig. 2k), since the temperature rise enhances the carrier scattering effect.

Considering that the carrier density variation of $\text{Bi}_2\text{O}_2\text{Se}$ is originated from the p-typed doping of the NO_2 adsorption, its thermodynamic feasibility can be justified

by analyzing the ionization energy of Bi₂O₂Se (ΔE_D). Herein, the ionization energy of Bi₂O₂Se is extracted by linearly fitting the value of $\ln(n_0)$ and $1/T$ when the operating temperature is low, obeying the formula below:

$$n_0 = \left(\frac{N_D N_C}{2}\right)^{\frac{1}{2}} \exp\left(-\frac{\Delta E_D}{2k_B T}\right) \quad (3)$$

where N_D and N_C are donor concentration and effective state density of conduction band, respectively. Then the temperature-dependent carrier-density variation is investigated for the Bi₂O₂Se in vacuum as the temperature decreases from 64 K to 50 K. Fig. 21 presents the linear fitting based on the values of $\ln(n_0)$ and $1/T$, then the slope is extracted as $-E_D/2k$ or $-E_D/k$. The ionization energy of 0.025 eV is acquired for the highly-doped character of the Bi₂O₂Se. By this low ionization energy and the highly-doped feature aforementioned, the p-type doping of the adsorbates is highly feasible to occur on the Bi₂O₂Se at room temperature.”

Q2. Figure 2 (e, h): Typically, in a device dominated by the thermionic emission (TE) mechanism, the device resistance should decrease as temperature increases. However, the behavior observed in Figure 2 (e, h) appears contradictory. Could you clarify the reason behind this?

Author reply: Thank the reviewer very much for this comment. In Figure 2 (e, h), the device resistance increases when the temperature increases from 280 K to 320 K, indicating the highly-doped characteristic of the Bi₂O₂Se which suppresses the effect of the Schottky barrier height (SBH) [Fu, H., Wu, J., Peng, H., Yan, B. Self-modulation doping effect in the high-mobility layered semiconductor Bi₂O₂Se. *Phys. Rev. B* **97**, 241203 (R) (2018).]. We have added this explanation into the related content and pasted it below for viewing convenient:

In section “**Fabrication of Schottky-Junction-based Gas Sensor**” of the manuscript: “As the temperature increases from 280 K to 320 K, the sample resistance increases, indicating the highly-doped characteristic of the Bi₂O₂Se which suppresses the effect of the Schottky barrier height (SBH)²⁷.”

Q3-1. Figure 2 (f, i): The method and analysis for extracting the Schottky Barrier Height (SBH) from Arrhenius plots seem incorrect. The negative slope of the plot should represent SBH, and accurate extraction of SBH requires suppression of thermionic emission effects, especially at lower temperatures. Furthermore, extracting SBH correctly is possible at the flat band V_g point.

Author reply: Many thanks for the reviewer for this comment. We have updated the content and revealed that the linearly fitting between $\ln(I/T^2)$ and T^{-1} is applied to extract the value of $(qV_{ds} - \Delta E)/k_B$. Then SBH value ΔE is obtained for the certain applied voltage V_{ds} . In this mean, the fitted slope value can be positive when the value of qV_{ds} was larger than the SBH value ΔE . Meanwhile, the operating temperature range is suitable for the SBH existence, by which the suppression of the thermionic emission effects endows the I - V curve with a non-linear relation in Figure 2e and h. The updated content is pasted below for reading convenience:

In section “**Fabrication of Schottky-Junction-based Gas Sensor**” of the manuscript: “The variation of SBH is estimated by fitting $\ln(I/T^2)$ and T^{-1} , according to the thermionic emission theory. In details, the thermionic emission theory describes the correlation between the current I and SBH (ΔE) as follows:

$$I = SA^*T^2 \exp\left(\frac{-\Delta E}{k_B T}\right) \left[\exp\left(\frac{qV_{ds}}{k_B T}\right) - 1\right] \quad (1)$$

where S , A^* , q , and k_B are respectively contact area, Richard coefficient, charge quantity, and Boltzmann constant. Note that when the applied voltage V_{ds} is large enough and the equation (1) can be simplified as equation (2) below:

$$I = SA^*T^2 \exp\left(\frac{qV_{ds} - \Delta E}{k_B T}\right) \quad (2)$$

where the slope extracted by the linearly fitting between $\ln(I/T^2)$ and T^{-1} is served as the value of $(qV_{ds} - \Delta E)/k_B$, and the SBH value ΔE is obtained with the certain applied voltage V_{ds} . In NO_2 , the ΔE (NO_2) is then estimated to be 2.795 eV (V_{ds} : 3 V) which is lower than that in vacuum (ΔE (Vac.): 2.846 eV). It is indicated that the adsorbed NO_2 decreases the carrier density of the $\text{Bi}_2\text{O}_2\text{Se}$ (n_0) and lowers $\text{Bi}_2\text{O}_2\text{Se}$ Fermi level, where the SBH reduces by 0.051 eV.”

Q3-2. The temperature range used in the current analysis seems too narrow and inadequate to exclude TE effects.

Author reply: Many thanks to the reviewer for this point. According to the reviewer's suggestion, the I - V relation is analyzed in a wide-range temperature ranging from 80 K to 360 K and the Schottky barrier height is extracted for confirming the existence of the Schottky junction barrier. The related addition is presented in S11 and pasted herein for reading convenience:

In S11 of the supplementary information:

“S11. The temperature-dependent I - V relation of the $\text{Bi}_2\text{O}_2\text{Se}$ sensor

To further characterize the Schottky junction barrier formed along the $\text{Bi}_2\text{O}_2\text{Se}$ -Au contact, the I - V relation is analyzed in a wide-range temperature ranging from 80 K to 360 K. Figure S10 presents the I - V curves for the sample at different temperatures (e.g., 80 K, 120 K, and 160 K), where the non-linear correlation exists for the I - V relation and indicates the existence of the Schottky junction barrier. To extract the Schottky barrier height, a linearly fitting between the $\ln(I/T^2)$ and T^{-1} is performed and the slope is the value of $(qV_{\text{ds}} - \Delta E)/k_{\text{B}}$, according to the thermionic emission theory. The inset in Figure S10 shows that, the Schottky barrier height is extracted to be 2.952 eV, confirming the formation of the Schottky junction barrier.

Fig. S10 | The temperature-dependent I - V correlation of the $\text{Bi}_2\text{O}_2\text{Se}$ sensor and its Schottky barrier height analysis (the inset).”

Q4. Figure 2 (j, k, l): The gas environment is not clearly stated (presumably vacuum). Also, for Figure 2 (l), consider revising the notation to $(1/T)$.

Author reply: We would like to highly appreciate the reviewer for this question and have updated the related. Please see the updated as follows for convenience:

In section “**Fabrication of Schottky-Junction-based Gas Sensor**” of the manuscript: “To reveal this carrier-density-dominated barrier variation, the $\text{Bi}_2\text{O}_2\text{Se}$ is fabricated into a Hall device for temperature-dependent resistance measurement (supplementary information S4). The temperature range is 2-380 K, and the test is conducted in Ar. Fig. 2j and Fig. S4c show that the resistance (R) is positively correlated to the temperature (T).”

“Then the temperature-dependent carrier-density variation is investigated for the $\text{Bi}_2\text{O}_2\text{Se}$ in vacuum as the temperature decreases from 64 K to 50 K.”

“

Fig. 2 | Adsorbate-Induced Variation of the Schottky-Junction Barrier. **a** The photograph of the Bi₂O₂Se gas sensor (scale bar: 12 μm). **b** AFM image showing topology of the Bi₂O₂Se-Au contact (scale bar: 3 μm). **c** The KPFM pattern of Bi₂O₂Se-Au contact (scale bar: 3 μm). **d** The energy band profile of the device in vacuum (vac.). **e** and **f** The temperature-dependent I - V plot of the device in vacuum and its linear fitting (**f**, $\ln(I/T^2)$ and T^{-1}). **g** The energy band profile of the device in NO₂ (700 ppt). **h** The I - V relation as the function of temperature in NO₂ (700 ppt). **i** The linear fitting between $\ln(I/T^2)$ and T^{-1} for I - V curve in **h**. **j** and **k** The temperature-dependent resistance (**j**) and mobility (**k**) in Ar environment. **l** The linear

fitting for extraction of ionization energy (the measurement conducted in vacuum).”

Q5. Reference: Please provide a reference for the inverse relationship between temperature and mobility at high doping concentrations, as observed in Fig 2j.

Author reply: Thank the reviewer very much for this suggestion. The related reference is added and related content is updated with more elaborations for clearly explaining the highly-doped feature of the Bi₂O₂Se. Please read the updated below for convenience:

(1) In section “**Fabrication of Schottky-Junction-based Gas Sensor**” of the manuscript:

“Fig. 2j and Fig. S4c show that the resistance (R) is positively correlated to the temperature (T). This metal-like R - T behavior indicates the highly-doped feature of the Bi₂O₂Se, where this highly-doped character is reported to be originated from the self-modulation doping effect²⁷. The highly-doped feature of the Bi₂O₂Se is further supported by the negative relation between the mobility of Bi₂O₂Se and temperature (Fig. 2k), since the temperature rise enhances the carrier scattering effect.”

(2) In the reference section of the manuscript:

“[27] Fu, H., Wu, J., Peng, H., Yan, B. Self-modulation doping effect in the high-mobility layered semiconductor Bi₂O₂Se. *Phys. Rev. B* **97**, 241203 (R) (2018).”

Q6. Explanation and Clarity in Figure 5: The description in Figure 5 is unclear. A more detailed and understandable explanation of the principle component and its implications is necessary. As Nature Communications is a general science journal, the explanation should be accessible to a broader audience.

Author reply: We would like to highly appreciate the reviewer for this suggestion and have updated the related content for a more detailed and understandable explanation of the principle component analysis. The related is pasted following for reading convenience:

In the section “**Molecularly Specific Detection towards Trace Gases**” of the

manuscript:

“To output molecular characteristic, this research conducts the principle component analysis based on the multiple sensing signals (Figure 5a and b). In this way, the sensing signals can be transferred into a new coordinate axis, where the component projected on each axis can be displayed with the maximum variation. It is therefore more distinct for the molecular characteristic presentation since the sensing signals for each molecule can be sufficiently dispersed. It means that the molecular characteristic can be differential by the well-dispersed component projected on each axis and applied for molecularly specific detection. In the principle component analysis, the responses of the multiple signals are firstly linearly fitted with the corresponding gas-concentration. The fitted parameters involving the slope and the intercept are obtained for different target gases (see Table S2 in supplementary information S9). These parameters are analyzed by the principle component analysis. Fig. 5a shows the three-dimensional component plot based on the multiple sensing signals. The projections on the principle component 1-3 are differential for each gas. These none-overlaid projections can be served as the molecule characteristics. In contrast, the projections are highly overlaid for the principle component analysis based on the fitted data of the resistance (*i.e.*, slope and intercept originated from the linear fitting), where only is two-dimensional plot presented in Fig. 5b. The reason for this fact is that, in the principle component analysis, the variance (α) originated from the multiple sensing signals is α^6 ($\alpha > 1$), which is 4 orders of magnitude larger than that of the traditional resistance. Thus the projections on the principle component 1-3 can be differential to show the molecular characteristics. In this mean, this study further applies the principle component analysis to output the molecular characteristic of NO₂ mixture. Fig. 5a presents that, each mixture is featured with specific molecular characteristic and adjacent to that of NO₂, by which the contained NO₂ can be detected.

Fig. 5| Analysis of Molecular Characteristics. **a** and **b** The principle component analysis based on the multiple signals (**a**; containing those of R , X , and Θ) and the R (**b**).”

Q7. Statement in Section S6: The statement, "the adsorbate with weaker dipole moment shows shorter binding length, indicating a larger quantity of charge," seems contradictory to typical understandings of dipole moments in molecules. Typically, a weaker dipole moment implies less charge separation, which might not necessarily correlate with shorter binding lengths or larger charge quantities.

A more detailed explanation or a simplified statement such as, "The charge induced on the surface by the dipole moment acts as a scattering center, thus changes in dipole moment can lead to changes in mobility," would be more appropriate.

Author reply: We would like to highly appreciate the reviewer for this comment. According to the reviewer’ suggested statement that “the charge induced on the surface by the dipole moment acts as a scattering center”, we have conducted the computation about the change in dipole moment of target molecule adsorbed. It is found that a negative correlation exists between the change in dipole moment and the mobility, confirming the statement suggested by the reviewer. Table S1 shows the dipole moment change for each adsorbed molecule, where the change of NO_2 adsorbed $\frac{P_{ads}-P_0}{P_0}$ is highest and results in the strongest scattering on the surface ($0 \leq \frac{P_{ads}-P_0}{P_0} \leq 1$). More detailed elaboration is presented in the updated version of

S6 and pasted herein for reading convenience:

In S6 of the supplementary information:

“S6. The Surface-Scattering-Dominated Sensing Mechanism

Figure S6 illustrates the surface-scattering-dominated sensing mechanism. The impedance angle is correlated with the change in dipole moment of the target molecule adsorbed $\frac{P_{ads.}-P_0}{P_0}$, where P_0 and $P_{ads.}$ are respectively dipole moments for the target gas and its adsorbed state. Firstly, the target molecule is adsorbed on the sensing block to induce two kinds of surficial phenomena, *i.e.*, surface scattering and adsorbate doping. The adsorbate doping varies the carrier density of the sensing block (n) to adjust its resistance (R). In this mean, the carrier-density-dominated sensing mechanism outputs the resistive signal but shows two issues: (1) the surface scattering usually interferes the resistive signal and degrades the sensitivity; (2) the molecular hybrid orbital energy cannot be distinguished based on the adsorbate doping at room temperature or the higher.

Herein, in the trace gas detection, the impedance angle is extracted as the sensing signal featured with molecular specificity. The reason for this fact is that the target molecule adsorbed forms as the scattering center. This scattering center is characterized by the charge quantity (σ), which is positively related to the change in the dipole moment of the target molecule ($\sigma \propto \frac{P_{ads.}-P_0}{P_0}$, $0 \leq \frac{P_{ads.}-P_0}{P_0} \leq 1$). In details, the target molecule is physically adsorbed on the $\text{Bi}_2\text{O}_2\text{Se}$ in the preliminary state, where its charge center is of charge quantity σ_0 and featured with dipole moment of P_0 . Next the surface doping occurs and indicates the change in dipole moment of the target molecule adsorbed $\frac{P_{ads.}-P_0}{P_0}$, where the adsorbed state of the molecule is of dipole moment of $P_{ads.}$. This adsorption induces the charge variation for the target molecule $\sigma_{ind.}$, and its value is equal to $\frac{P_{ads.}-P_0}{P_0}\sigma_0$. Equivalent to the parallel-plate capacitance, the electric field E_0 in the preliminary state turns into the adsorbed state $E_{ads.}$, which is featured with the permittivity $\varepsilon_{r,ads.}$. This permittivity $\varepsilon_{r,ads.}$ is positively correlated to the charge of scattering center σ . Then the charge of scattering

center σ is confirmed to be positively related to the change in the dipole moment of the target molecule ($\sigma \propto \frac{P_{ads.}-P_0}{P_0}$, $0 \leq \frac{P_{ads.}-P_0}{P_0} \leq 1$).

In the case of ionized impurity scattering, the carrier mobility is related to the charge quantity. Considering that the **impedance** angle is mainly dominated by the carrier mobility (μ) in the gas sensing, the **impedance** angle is then dependent on **the change in dipole moment of the target molecule adsorbed** and featured with molecular specificity.

Fig. S6 | The schematic illustrating the surface-scattering-dominated sensing mechanism. Compared against the traditional mechanism (outputting the resistive signal), the surface-scattering mechanism is featured with high specificity.

Regarding the target gases, Table S1 shows their molecular features such as dipole moment and the highest occupied molecular orbital (HOMO) level. Mulliken charge transferring is extracted for each target gas to clarify the adsorbate doping, and the surface doping can be classified into the n-type and the p-type doping. Quantum mechanical computations are based on the Gaussian 09 computational suite, obeying theory of Coupled Cluster Singles, Doubles, and Perturbative Triples; and the correlation-consistent polarized valence triple-zeta basis set is adopted for a fine balance between computational feasibility and accuracy⁷⁻¹⁰. Additionally, the PBE0 density functional method complemented with the D3(BJ) dispersion correction is employed to conduct all computational analyses. Established by Adamo and Barone in their foundational work on the PBE0 model, this strategy is applied for ensuring accurate and reliable density functional methods without parameter adjustment¹¹⁻¹². Noted for its precise Coulomb-fitting for elements ranging from hydrogen to radon¹³, the def2-SVP basis set is utilized for atoms in the geometry optimizations phase. This basis set choice is subjected to its established accuracy and efficiency. The restricted optimizations of the crystal structure are conducted and crucial for the investigation. Unrestricted optimizations were performed for specific molecules, including nitrogen dioxide, isopropanol, formaldehyde, methanol, acetone, oxygen, ammonia, ethane, ethanol, acetic acid, formic acid, and acetaldehyde. By the unique properties and behaviors of these molecules, the distinction in methodology shows a thorough and nuanced analysis. Guided by the principles laid out in the aforementioned works, the primary research is focused on the adsorption studies on crystals and small molecules, leveraging the combination of the PBE0 density functional method and the def2-SVP basis set.

Considering that the certain value of $\frac{P_{ads}-P_0}{P_0}$ ($0 \leq \frac{P_{ads}-P_0}{P_0} \leq 1$) is accessible for the surface-scattering-dominated sensing mechanism, the NO₂ is characterized with the highest value of $\frac{P_{ads}-P_0}{P_0}$ (0.07), by which the NO₂ is estimated to form a scattering center to reduce the mobility and enhance the impedance angle at the largest level.

Table S1. The Molecular Features of the Target Gas and Its Adsorbed State

Target molecule	P_0 (Debye)	$P_{ads.}$ (Debye)	$\frac{P_{ads.} - P_0}{P_0}$	HOMO (eV)	Muliken charge transferring
O ₂	0	0	0	-13.16	-0.546
CH ₃ COOH	4.79	4.22	-0.12	-12.27	-0.045
CH ₃ COCH ₃	3.23	2.93	-0.09	-11.18	-0.08
CH ₂ O	2.82	2.49	-0.12	-12.02	-0.085
CH ₃ OH	1.73	1.76	0.02	-12.08	0.21
CH ₃ CH ₂ OH	1.70	1.71	0.01	-12.00	0.215
CHOOH	4.38	3.97	-0.09	-12.70	-0.049
CH ₃ CHOHCH ₃	1.72	1.63	-0.05	-11.86	0.223
NH ₃	1.69	1.55	-0.08	-11.67	0.393
NO ₂	0.45	0.48	0.07	-13.22	-0.259
CH ₃ CH ₂ CHO	3.27	2.60	-0.20	-11.40	-0.099
CH ₃ CH ₃	0	0.32	-	-13.25	0.373

”

Q8. Inconsistency in Section S6: There is a contradiction in stating that "The charge quantity is positively related to the binding length between the target molecule and the sensing block." but concluding that a lower dipole results in a higher charge. This needs clarification or correction.

Author reply: Many thanks for the reviewer for this point, and we have made a correction accordingly. In this correction, the charge quantity is described to be positively related to the change in dipole moment of the target molecule adsorbed. More details are pasted below for viewing convenience:

In S6 of the supplementary information:

“Herein, in the trace gas detection, the impedance angle is extracted as the sensing signal featured with molecular specificity. The reason for this fact is that the target molecule adsorbed forms as the scattering center. This scattering center is

characterized by the charge quantity (σ), which is positively related to the change in the dipole moment of the target molecule ($\sigma \propto \frac{P_{ads.-P_0}}{P_0}$, $0 \leq \frac{P_{ads.-P_0}}{P_0} \leq 1$). In details, the target molecule is physically adsorbed on the Bi₂O₂Se in the preliminary state, where its charge center is of charge quantity σ_0 and featured with dipole moment of P_0 . Next the surface doping occurs and indicates the change in dipole moment of the target molecule adsorbed $\frac{P_{ads.-P_0}}{P_0}$, where the adsorbed state of the molecule is of dipole moment of $P_{ads.}$. This adsorption induces the charge variation for the target molecule $\sigma_{ind.}$, and its value is equal to $\frac{P_{ads.-P_0}}{P_0}\sigma_0$. Equivalent to the parallel-plate capacitance, the electric field E_0 in the preliminary state turns into the adsorbed state $E_{ads.}$, which is featured with the permittivity $\epsilon_{r,ads.}$. This permittivity $\epsilon_{r,ads.}$ is positively correlated to the charge of scattering center σ . Then the charge of scattering center σ is confirmed to be positively related to the change in the dipole moment of the target molecule ($\sigma \propto \frac{P_{ads.-P_0}}{P_0}$, $0 \leq \frac{P_{ads.-P_0}}{P_0} \leq 1$).

In the case of ionized impurity scattering, the carrier mobility is related to the charge quantity. Considering that the impedance angle is mainly dominated by the carrier mobility (μ) in the gas sensing, the impedance angle is then dependent on the change in dipole moment of the target molecule adsorbed and featured with molecular specificity.

Fig. S6 | The schematic illustrating the surface-scattering-dominated sensing mechanism. Compared against the traditional mechanism (outputting the resistive signal), the surface-scattering mechanism is featured with high specificity.

Regarding the target gases, Table S1 shows their molecular features such as dipole moment and the highest occupied molecular orbital (HOMO) level. Quantum mechanical computations are based on the Gaussian 09 computational suite, obeying theory of Coupled Cluster Singles, Doubles, and Perturbative Triples; and the correlation-consistent polarized valence triple-zeta basis set is adopted for a fine balance between computational feasibility and accuracy⁷⁻¹⁰.

Considering that the certain value of $\frac{P_{ads.}-P_0}{P_0}$ ($0 \leq \frac{P_{ads.}-P_0}{P_0} \leq 1$) is accessible for the surface-scattering-dominated sensing mechanism, the NO₂ is characterized with the highest value of $\frac{P_{ads.}-P_0}{P_0}$ (0.07). Together with the low HOMO at -13.22 eV, the NO₂ is estimated to form a scattering center to reduce the mobility and enhance the impedance angle at the largest level.

Table S1. The Molecular Features of the Target Gas and Its Adsorbed State

Target molecule	P_0 (Debye)	$P_{ads.}$ (Debye)	$\frac{P_{ads.} - P_0}{P_0}$	HOMO (eV)
O ₂	0	0	0	-13.16
CH ₃ COOH	4.79	4.22	-0.12	-12.27
CH ₃ COCH ₃	3.23	2.93	-0.09	-11.18
CH ₂ O	2.82	2.49	-0.12	-12.02
CH ₃ OH	1.73	1.76	0.02	-12.08
CH ₃ CH ₂ OH	1.70	1.71	0.01	-12.00
CHOOH	4.38	3.97	-0.09	-12.70
CH ₃ CHOHCH ₃	1.72	1.63	-0.05	-11.86
NH ₃	1.69	1.55	-0.08	-11.67
NO ₂	0.45	0.48	0.07	-13.22
CH ₃ CH ₂ CHO	3.27	2.60	-0.20	-11.40
CH ₃ CH ₃	0	0.32	-	-13.25

”

Q9. In page 4, line 117, Please revise the expression.

Figure 1a shows the high angle annular dark-field scanning transition microscopy (HAADF-STEM) image of the sample from the cross-sectional view.

Author reply: Many thanks to the reviewer for this point, and the related content is revised and pasted following for reading convenience:

In the section “**Two-Dimensional Bi₂O₂Se with Atomically-Flat Surface**” of the manuscript:

“Figure 1a shows the high angle annular dark-field scanning transition **electron** microscopy (HAADF-STEM) image of the sample **in** the cross-sectional view.”

Q10. In page 6, line 150, Please revise the expression.

“The *I-V* characterization is conducted in vacuum/700 ppt NO₂ at different temperature (*T*, ranging in 280-310 K)”

Author reply: Many thanks for the reviewer for this question. We have revised the related content and pasted following for reading convenience:

In the section “**Fabrication of Schottky-Junction-based Gas Sensor**” of the manuscript:

“The *I-V* characterizations are performed on the Bi₂O₂Se-Au contact at different temperatures (*T*) ranging from 280 K to 310 K, and these measurements are respectively conducted in vacuum and in 700 ppt NO₂.”

Q11. Bi₂O₂Se is a promising material for various gas sensing reactions, as can be seen in the paper "One-Step Synthesis of Bi₂O₂Se Microstructures for Trace Oxygen Gas Sensor Application" by Ji kwon Bae, Hyeon Ho Cho, et al. published in *Sensors & Actuators: B* (2023). indicates that Therefore, the author recommends adding an application for Bi₂O₂Se by referring to various papers (https://www.sciencedirect.com/science/article/pii/S0925400523011139?casa_token=l_xretiByKswAAAAA:ltY_o02GI08liIJvPoDXk7m1ePai8bItU7IR2vJflPBEPvnwdKs-wPOM_hmXcdkZ5Yjx_sJs) This study is a single Bi₂O₂Se microstructures are successfully synthesized using a zoned chemical vapor deposition (CVD) method,

demonstrating their effectiveness as O₂ sensors.

Author reply: Thank the reviewer very much for this suggestion. We have added the related references about the Bi₂O₂Se-based gas sensing and updated the content accordingly.

Please see the updated below for convenience:

(1) In the section “**Two-Dimensional Bi₂O₂Se with Atomically-Flat Surface**” of the manuscript:

“Considering that Bi₂O₂Se emerges as a promising material for gas sensing^{25,26}, this investigation selects the Bi₂O₂Se as the sensing block. By molecular beam epitaxy, this research is to synthesize single crystalline of Bi₂O₂Se with atomic roughness on the SrTiO₃ (001) substrate.”

(2) In the reference section of the manuscript:

“[25] Bae, J. K., Cho, H. H., Shin, H. et al., One-step synthesis of Bi₂O₂Se microstructures for trace oxygen gas sensor application, *Sens. Actuators B Chem.* **394**, 134398 (2023).

[26] Xu, S., Fu, H., Tian, Y. et al., Exploiting two-dimensional Bi₂O₂Se for trace oxygen detection. *Angew. Chem. Int. Ed.* **59**, 17938 (2020)”

Q12. One of the important properties of a gas sensor is its stability. The gas sensor should ensure stability over a long period of time even with various times of gas exposure. The authors should supplement with additional data on stability.

Author reply: Thank the reviewer very much for this suggestion. We have added stability test in the supplementary information, where the acetone detection is conducted after 20 days. The multiple sensing signals remain almost constant as compared against those of the original. It is revealed that the gas sensor is of a comparable stability. For viewing convenience, the related addition is pasted as follows:

In S12 of the supplementary information:

“**S12. The stability of the Bi₂O₂Se sensor**

In Figure S11, the Bi₂O₂Se sensor is applied for acetone detection and the original response is compared against that after 20 days. In the detection towards 7 ppm acetone after 20 days, the sensing responses remain almost constant for the multiple signals such as the resistance, the reactance, and the impedance angle. The most degradation occurs for the resistance responsivity, manifesting by around 10 % reduction as compared to the original. Regarding the reactance and the impedance angle signal, the negligible degradation exists for their responsivities. Then the sensor is ensured with the comparable stability over a long period of time.

Fig. S11 | The stability test for the sensor after 20 days. The target gas is acetone (ace.) and its concentration is 7 ppm. Multiple sensing signals are collected, including the resistance, the reactance, and the impedance angle.”

Q13. In Figure 2, the author set the temperature range to 2-300 K to explain resistance according to temperature. However, correlations should be investigated not only at

temperatures below the room temperature but also at temperatures above the room temperature.

Author reply: We would like to highly appreciate the reviewer for this suggestion. The temperature-dependent resistance above the room temperature is added in SI, where the resistance is positively correlated with the temperature. This correlation confirms the metal-like character of the $\text{Bi}_2\text{O}_2\text{Se}$ mentioned in the manuscript. Please see the added characterization pasted below for convenience:

In S4 of the supplementary information:

“S4. The $\text{Bi}_2\text{O}_2\text{Se}$ Hall Device

Figure S4a shows the atomic force microscopy (AFM) image of the $\text{Bi}_2\text{O}_2\text{Se}$ sheet, where its thickness is 7 nm and it is applied for the Hall device fabrication. Fig. S4b presents the SEM image of the $\text{Bi}_2\text{O}_2\text{Se}$ Hall device. The $\text{Bi}_2\text{O}_2\text{Se}$ sheet is bonded with six electrodes for the Hall test. The electrode is consisted of 10-nm-thin Pd underneath and 90-nm-thin Au on the top. The R - T characterization is conducted on the $\text{Bi}_2\text{O}_2\text{Se}$ device when the temperature increases from 10 K to 380K. Fig. S4c shows a positive relation between the $\text{Bi}_2\text{O}_2\text{Se}$ resistance and temperature, indicating the metal-like feature of the $\text{Bi}_2\text{O}_2\text{Se}$.

Fig. S4| a The AFM image of the $\text{Bi}_2\text{O}_2\text{Se}$ sheet (scale bar: 3 μm). **b** The SEM image of the $\text{Bi}_2\text{O}_2\text{Se}$ device (scale bar: 2 μm). **c** R - T characterization for the $\text{Bi}_2\text{O}_2\text{Se}$ device.”

- Recommendations for Strengthening the Manuscript:

R1. Demonstrating the change in mobility due to the dipole moment through

calculations could significantly enhance the persuasiveness of the paper.

Author reply: Many thanks for the reviewer for this suggestion. In my former investigation into the carrier-mobility-dominated gas sensing, the dipole moment of the target gas is demonstrated to be correlated with the carrier mobility [Xu, S., Zhao, H., Xu, Y., Xu, R., Lei, Y. Carrier mobility-dominated gas sensing: a room-temperature gas-sensing mode for SnO₂ nanorod array sensors. *ACS Appl. Mater. Interfaces* **10**, 13895 (2018).]. We have added this reference into the manuscript to enhance the persuasiveness of the paper. Additionally, according to the reviewer's suggestion, we have conducted the calculation about the dipole-moment change for the molecule adsorbed, and proved that the change in dipole moment of the target molecule adsorbed is negatively related to the carrier mobility. The main updated is pasted herein for reading convenience:

(1) In section “**Fabrication of Schottky-Junction-based Gas Sensor**” of manuscript: “Note that the resistance in NO₂ is higher than that in vacuum, indicating the p-type doping/surface scattering originated from the adsorbed NO₂²⁸.”

(2) In section “**Molecularly Specific Detection towards Trace Gases**” of the manuscript:

“The ppt-response of impedance angle is attributed to the apparent change in dipole moment of the NO₂ adsorbed, which is positively related to the surface scattering (supplementary information S6).”

(3) In the discussion section of the manuscript:

“The molecular selectivity is proved to be correlated with the change in dipole moment of target molecules adsorbed and manifests as the impedance angle.”

(4) In the reference section of the manuscript:

“[28] Xu, S., Zhao, H., Xu, Y., Xu, R., Lei, Y. Carrier mobility-dominated gas sensing: a room-temperature gas-sensing mode for SnO₂ nanorod array sensors. *ACS Appl. Mater. Interfaces* **10**, 13895 (2018).”

(2) In S6 of the supplementary information:

“S6. The Surface-Scattering-Dominated Sensing Mechanism

Figure S6 illustrates the surface-scattering-dominated sensing mechanism. The impedance angle is correlated with the change in dipole moment of the target molecule adsorbed $\frac{P_{ads.}-P_0}{P_0}$, where P_0 and $P_{ads.}$ are respectively dipole moments for the target gas and its adsorbed state. Firstly, the target molecule is adsorbed on the sensing block to induce two kinds of surficial phenomena, *i.e.*, surface scattering and adsorbate doping. The adsorbate doping varies the carrier density of the sensing block (n) to adjust its resistance (R). In this mean, the carrier-density-dominated sensing mechanism outputs the resistive signal but shows two issues: (1) the surface scattering usually interferes the resistive signal and degrades the sensitivity; (2) the molecular hybrid orbital energy cannot be distinguished based on the adsorbate doping at room temperature or the higher.

Herein, in the trace gas detection, the impedance angle is extracted as the sensing signal featured with molecular specificity. The reason for this fact is that the target molecule adsorbed forms as the scattering center. This scattering center is characterized by the charge quantity (σ), which is positively related to the change in the dipole moment of the target molecule ($\sigma \propto \frac{P_{ads.}-P_0}{P_0}$, $0 \leq \frac{P_{ads.}-P_0}{P_0} \leq 1$). In details, the target molecule is physically adsorbed on the $\text{Bi}_2\text{O}_2\text{Se}$ in the preliminary state, where its charge center is of charge quantity σ_0 and featured with dipole moment of P_0 . Next the surface doping occurs and indicates the change in dipole moment of the target molecule adsorbed $\frac{P_{ads.}-P_0}{P_0}$, where the adsorbed state of the molecule is of dipole moment of $P_{ads.}$. This adsorption induces the charge variation for the target molecule $\sigma_{ind.}$, and its value is equal to $\frac{P_{ads.}-P_0}{P_0}\sigma_0$. Equivalent to the parallel-plate capacitance, the electric field E_0 in the preliminary state turns into the adsorbed state $E_{ads.}$, which is featured with the permittivity $\varepsilon_{r,ads.}$. This permittivity $\varepsilon_{r,ads.}$ is positively correlated to the charge of scattering center σ . Then the charge of scattering

center σ is confirmed to be positively related to the change in the dipole moment of the target molecule ($\sigma \propto \frac{P_{ads.}-P_0}{P_0}$, $0 \leq \frac{P_{ads.}-P_0}{P_0} \leq 1$).

In the case of ionized impurity scattering, the carrier mobility is related to the charge quantity. Considering that the **impedance** angle is mainly dominated by the carrier mobility (μ) in the gas sensing, the **impedance** angle is then dependent on **the change in dipole moment of the target molecule adsorbed** and featured with molecular specificity.

Fig. S6 | The schematic illustrating the surface-scattering-dominated sensing mechanism. Compared against the traditional mechanism (outputting the resistive signal), the surface-scattering mechanism is featured with high specificity.

Regarding the target gases, Table S1 shows their molecular features such as dipole moment and the highest occupied molecular orbital (HOMO) level. Mulliken charge transferring is extracted for each target gas to clarify the adsorbate doping, and the surface doping can be classified into the n-type and the p-type doping. Quantum mechanical computations are based on the Gaussian 09 computational suite, obeying theory of Coupled Cluster Singles, Doubles, and Perturbative Triples; and the correlation-consistent polarized valence triple-zeta basis set is adopted for a fine balance between computational feasibility and accuracy⁷⁻¹⁰. Additionally, the PBE0 density functional method complemented with the D3(BJ) dispersion correction is employed to conduct all computational analyses. Established by Adamo and Barone in their foundational work on the PBE0 model, this strategy is applied for ensuring accurate and reliable density functional methods without parameter adjustment¹¹⁻¹². Noted for its precise Coulomb-fitting for elements ranging from hydrogen to radon¹³, the def2-SVP basis set is utilized for atoms in the geometry optimizations phase. This basis set choice is subjected to its established accuracy and efficiency. The restricted optimizations of the crystal structure are conducted and crucial for the investigation. Unrestricted optimizations were performed for specific molecules, including nitrogen dioxide, isopropanol, formaldehyde, methanol, acetone, oxygen, ammonia, ethane, ethanol, acetic acid, formic acid, and acetaldehyde. By the unique properties and behaviors of these molecules, the distinction in methodology shows a thorough and nuanced analysis. Guided by the principles laid out in the aforementioned works, the primary research is focused on the adsorption studies on crystals and small molecules, leveraging the combination of the PBE0 density functional method and the def2-SVP basis set.

Considering that the certain value of $\frac{P_{ads.-P_0}}{P_0}$ ($0 \leq \frac{P_{ads.-P_0}}{P_0} \leq 1$) is accessible for the surface-scattering-dominated sensing mechanism, the NO₂ is characterized with the highest value of $\frac{P_{ads.-P_0}}{P_0}$ (0.07), by which the NO₂ is estimated to form a scattering center to reduce the mobility and enhance the impedance angle at the largest level.

Table S1. The Molecular Features of the Target Gas and Its Adsorbed State

Target molecule	P_0 (Debye)	$P_{ads.}$ (Debye)	$\frac{P_{ads.} - P_0}{P_0}$	HOMO (eV)	Muliken charge transferring
O ₂	0	0	0	-13.16	-0.546
CH ₃ COOH	4.79	4.22	-0.12	-12.27	-0.045
CH ₃ COCH ₃	3.23	2.93	-0.09	-11.18	-0.08
CH ₂ O	2.82	2.49	-0.12	-12.02	-0.085
CH ₃ OH	1.73	1.76	0.02	-12.08	0.21
CH ₃ CH ₂ OH	1.70	1.71	0.01	-12.00	0.215
CHOOH	4.38	3.97	-0.09	-12.70	-0.049
CH ₃ CHOHCH ₃	1.72	1.63	-0.05	-11.86	0.223
NH ₃	1.69	1.55	-0.08	-11.67	0.393
NO ₂	0.45	0.48	0.07	-13.22	-0.259
CH ₃ CH ₂ CHO	3.27	2.60	-0.20	-11.40	-0.099
CH ₃ CH ₃	0	0.32	-	-13.25	0.373

”

R2. Given that *Nature Communications* covers multidisciplinary science, a more detailed and easily understandable explanation throughout the manuscript would be beneficial.

Author reply: Thank the reviewer very much for this suggestion. We have updated the manuscript for presenting a more detailed and easily understandable explanation. Some of the related are pasted herein for reading convenience:

(1) In “**Fabrication of Schottky-Junction-based Gas Sensor**” section of the:

“The variation of SBH is estimated by fitting $\ln(I/T^2)$ and T^{-1} , according to the thermionic emission theory. In details, the thermionic emission theory describes the correlation between the current I and SBH (ΔE) as follows:

$$I = SA^*T^2 \exp\left(\frac{-\Delta E}{k_B T}\right) \left[\exp\left(\frac{qV_{ds}}{k_B T}\right) - 1\right] \quad (1)$$

where S , A^* , q , and k_B are respectively contact area, Richard coefficient, charge

quantity, and Boltzmann constant. Note that when the applied voltage V_{ds} is large enough and the equation (1) can be simplified as equation (2) below:

$$I = SA^* T^2 \exp\left(\frac{qV_{ds} - \Delta E}{k_B T}\right) \quad (2)$$

where the slope extracted by the linearly fitting between $\ln(I/T^2)$ and T^{-1} is served as the value of $(qV_{ds} - \Delta E)/k_B$, and the SBH value ΔE is obtained with the certain applied voltage V_{ds} . In NO_2 , the ΔE (NO_2) is then estimated to be 2.795 eV (V_{ds} : 3 V) which is lower than that in vacuum (ΔE (Vac.): 2.846 eV). It is indicated that the adsorbed NO_2 decreases the carrier density of the $\text{Bi}_2\text{O}_2\text{Se}$ (n_0) and lowers $\text{Bi}_2\text{O}_2\text{Se}$ Fermi level, where the SBH reduces by 0.051 eV.

To reveal this carrier-density-dominated barrier variation, the $\text{Bi}_2\text{O}_2\text{Se}$ is fabricated into a Hall device for temperature-dependent resistance measurement (supplementary information S4). The temperature range is 2-380 K, and the test is conducted in Ar. Fig. 2j and Fig. S4c show that the resistance (R) is positively correlated to the temperature (T). This metal-like R - T behavior indicates the highly-doped feature of the $\text{Bi}_2\text{O}_2\text{Se}$, where this highly-doped character is reported to be originated from the self-modulation doping effect²⁷. The highly-doped feature of the $\text{Bi}_2\text{O}_2\text{Se}$ is further supported by the negative relation between the mobility of $\text{Bi}_2\text{O}_2\text{Se}$ and temperature (Fig. 2k), since the temperature rise enhances the carrier scattering effect.

Considering that the carrier density variation of $\text{Bi}_2\text{O}_2\text{Se}$ is originated from the p-typed doping of the NO_2 adsorption, its thermodynamic feasibility can be justified by analyzing the ionization energy of $\text{Bi}_2\text{O}_2\text{Se}$ (ΔE_D). Herein, the ionization energy of $\text{Bi}_2\text{O}_2\text{Se}$ is extracted by linearly fitting the value of $\ln(n_0)$ and $1/T$ when the operating temperature is low, obeying the formula below:

$$n_0 = \left(\frac{N_D N_C}{2}\right)^{\frac{1}{2}} \exp\left(-\frac{\Delta E_D}{2k_B T}\right) \quad (3)$$

where N_D and N_C are donor concentration and effective state density of conduction band, respectively. Then the temperature-dependent carrier-density variation is investigated for the $\text{Bi}_2\text{O}_2\text{Se}$ in vacuum as the temperature decreases from 64 K to 50 K. Fig. 2l presents the linear fitting based on the values of $\ln(n_0)$ and $1/T$, the slope is then extracted as $-E_D/2k$ or $-E_D/k$. The ionization energy of 0.025 eV is acquired for

the highly-doped character of the Bi₂O₂Se. By this low ionization energy and the highly-doped feature aforementioned, the p-type doping of the adsorbates is highly feasible to occur on the Bi₂O₂Se at room temperature.”

(2) In the section “**Molecularly Specific Detection towards Trace Gases**” of the manuscript:

“To output molecular characteristic, this research conducts the principle component analysis based on the multiple sensing signals (Figure 5a and b). In this way, the sensing signals can be transferred into a new coordinate axis, where the component projected on each axis can be displayed with the maximum variation. It is therefore more distinct for the molecular characteristic presentation since the sensing signals for each molecule can be sufficiently dispersed. It means that the molecular characteristic can be differential by the well-dispersed component projected on each axis and applied for molecularly specific detection. In the principle component analysis, the responses of the multiple signals are firstly linearly fitted with the corresponding gas-concentration. The fitted parameters involving the slope and the intercept are obtained for different target gases (see Table S2 in supplementary information S9). These parameters are analyzed by the principle component analysis. Fig. 5a shows the three-dimensional component plot based on the multiple sensing signals. The projections on the principle component 1-3 are differential for each gas. These none-overlaid projections can be served as the molecule characteristics. In contrast, the projections are highly overlaid for the principle component analysis based on the fitted data of the resistance (*i.e.*, slope and intercept originated from the linear fitting), where only is two-dimensional plot presented in Fig. 5b. The reason for this fact is that, in the principle component analysis, the variance (a) originated from the multiple sensing signals is a^6 ($a > 1$), which is 4 orders of magnitude larger than that of the traditional resistance. Thus the projections on the principle component 1-3 can be differential to show the molecular characteristics. In this mean, this study further applies the principle component analysis to output the molecular characteristic of NO₂ mixture. Fig. 5a presents that, each mixture is featured with specific molecular

characteristic and adjacent to that of NO₂, by which the contained NO₂ can be detected.

Fig. 5| Analysis of Molecular Characteristics. **a** and **b** The principle component analysis based on the multiple signals (**a**; containing those of R , X , and Θ) and the R (**b**).”

(3) In S6 of supplementary information:

“S6. The Surface-Scattering-Dominated Sensing Mechanism

Figure S6 illustrates the surface-scattering-dominated sensing mechanism. The impedance angle is correlated with the change in dipole moment of the target molecule adsorbed $\frac{P_{ads.}-P_0}{P_0}$, where P_0 and $P_{ads.}$ are respectively dipole moments for the target gas and its adsorbed state. Firstly, the target molecule is adsorbed on the sensing block to induce two kinds of surficial phenomena, *i.e.*, surface scattering and adsorbate doping. The adsorbate doping varies the carrier density of the sensing block (n) to adjust its resistance (R). In this mean, the carrier-density-dominated sensing mechanism outputs the resistive signal but shows two issues: (1) the surface scattering usually interferes the resistive signal and degrades the sensitivity; (2) the molecular hybrid orbital energy cannot be distinguished based on the adsorbate doping at room temperature or the higher.

Herein, in the trace gas detection, the impedance angle is extracted as the sensing signal featured with molecular specificity. The reason for this fact is that the target

molecule adsorbed forms as the scattering center. This scattering center is characterized by the charge quantity (σ), which is positively related to the change in the dipole moment of the target molecule ($\sigma \propto \frac{P_{ads.-P_0}}{P_0}$, $0 \leq \frac{P_{ads.-P_0}}{P_0} \leq 1$). In details, the target molecule is physically adsorbed on the Bi₂O₂Se in the preliminary state, where its charge center is of charge quantity σ_0 and featured with dipole moment of P_0 . Next the surface doping occurs and indicates the change in dipole moment of the target molecule adsorbed $\frac{P_{ads.-P_0}}{P_0}$, where the adsorbed state of the molecule is of dipole moment of $P_{ads.}$. This adsorption induces the charge variation for the target molecule $\sigma_{ind.}$, and its value is equal to $\frac{P_{ads.-P_0}}{P_0}\sigma_0$. Equivalent to the parallel-plate capacitance, the electric field E_0 in the preliminary state turns into the adsorbed state $E_{ads.}$, which is featured with the permittivity $\epsilon_{r,ads.}$. This permittivity $\epsilon_{r,ads.}$ is positively correlated to the charge of scattering center σ . Then the charge of scattering center σ is confirmed to be positively related to the change in the dipole moment of the target molecule ($\sigma \propto \frac{P_{ads.-P_0}}{P_0}$, $0 \leq \frac{P_{ads.-P_0}}{P_0} \leq 1$).

In the case of ionized impurity scattering, the carrier mobility is related to the charge quantity. Considering that the impedance angle is mainly dominated by the carrier mobility (μ) in the gas sensing, the impedance angle is then dependent on the change in dipole moment of the target molecule adsorbed and featured with molecular specificity.

Fig. S6 | The schematic illustrating the surface-scattering-dominated sensing mechanism. Compared against the traditional mechanism (outputting the resistive signal), the surface-scattering mechanism is featured with high specificity.

Regarding the target gases, Table S1 shows their molecular features such as dipole moment and the highest occupied molecular orbital (HOMO) level. Mulliken charge transferring is extracted for each target gas to clarify the adsorbate doping, and the surface doping can be classified into the n-type and the p-type doping. Quantum mechanical computations are based on the Gaussian 09 computational suite, obeying theory of Coupled Cluster Singles, Doubles, and Perturbative Triples; and the correlation-consistent polarized valence triple-zeta basis set is adopted for a fine balance between computational feasibility and accuracy⁷⁻¹⁰. Additionally, the PBE0 density functional method complemented with the D3(BJ) dispersion correction is employed to conduct all computational analyses. Established by Adamo and Barone in their foundational work on the PBE0 model, this strategy is applied for ensuring accurate and reliable density functional methods without parameter adjustment¹¹⁻¹². Noted for its precise Coulomb-fitting for elements ranging from hydrogen to radon¹³, the def2-SVP basis set is utilized for atoms in the geometry optimizations phase. This basis set choice is subjected to its established accuracy and efficiency. The restricted optimizations of the crystal structure are conducted and crucial for the investigation. Unrestricted optimizations were performed for specific molecules, including nitrogen dioxide, isopropanol, formaldehyde, methanol, acetone, oxygen, ammonia, ethane, ethanol, acetic acid, formic acid, and acetaldehyde. By the unique properties and behaviors of these molecules, the distinction in methodology shows a thorough and nuanced analysis. Guided by the principles laid out in the aforementioned works, the primary research is focused on the adsorption studies on crystals and small molecules, leveraging the combination of the PBE0 density functional method and the def2-SVP basis set.

Considering that the certain value of $\frac{P_{ads}-P_0}{P_0}$ ($0 \leq \frac{P_{ads}-P_0}{P_0} \leq 1$) is accessible for the surface-scattering-dominated sensing mechanism, the NO₂ is characterized with the highest value of $\frac{P_{ads}-P_0}{P_0}$ (0.07), by which the NO₂ is estimated to form a scattering center to reduce the mobility and enhance the impedance angle at the largest level.

Table S1. The Molecular Features of the Target Gas and Its Adsorbed State

Target molecule	P_0 (Debye)	$P_{ads.}$ (Debye)	$\frac{P_{ads.} - P_0}{P_0}$	HOMO (eV)	Muliken charge transferring
O ₂	0	0	0	-13.16	-0.546
CH ₃ COOH	4.79	4.22	-0.12	-12.27	-0.045
CH ₃ COCH ₃	3.23	2.93	-0.09	-11.18	-0.08
CH ₂ O	2.82	2.49	-0.12	-12.02	-0.085
CH ₃ OH	1.73	1.76	0.02	-12.08	0.21
CH ₃ CH ₂ OH	1.70	1.71	0.01	-12.00	0.215
CHOOH	4.38	3.97	-0.09	-12.70	-0.049
CH ₃ CHOHCH ₃	1.72	1.63	-0.05	-11.86	0.223
NH ₃	1.69	1.55	-0.08	-11.67	0.393
NO ₂	0.45	0.48	0.07	-13.22	-0.259
CH ₃ CH ₂ CHO	3.27	2.60	-0.20	-11.40	-0.099
CH ₃ CH ₃	0	0.32	-	-13.25	0.373

”

Reviewer #2 (Remarks to the Author):

The manuscript entitled "Molecularly Specific Detection towards Trace Nitrogen Dioxide by Utilizing Schottky-Junction-based Gas Sensor" proposes a crucial issue in the current gas sensor research, i.e., the lack of highly specific sensing mechanism for the semiconductor-based chemiresistive and chemical field-effect transistor sensors, which leads to the poor selectivity of the sensor toward specific target gas. Many problems existed in the manuscript and my comments are as follows:

Author reply: Thank the reviewer very much for the comments. We have point-by-point response these comments and updated the manuscript accordingly. The author reply is pasted below for reading convenience.

1-1. In the Introduction part, “NO₂ molecules firstly react with these oxygen ions to release the trapped electrons back to the sensing block.” This description is absolutely wrong, since NO₂ is an oxidizing gas.

Author reply: Many thanks to the reviewer for this comment. In the oxygen ionic model, oxygen molecules adsorbed can trap the electrons from the ionic oxide to form the oxygen ions [Huang, X., Meng, F., Pi, Z., Xu, W., Liu, J. Gas sensing behavior of a single tin dioxide sensor under dynamic temperature modulation. *Sens. Actuators B Chem.* **99**, 444 (2004).]; **these oxygen ions are of reducing feature and feasible to react with the oxidizing gas of NO₂**, resulting in the releasing of the trapped electron back to the sensing block.

We have updated the related introduction to clear illustrate the oxygen ionic model. Please read the updated content below:

(1) In the introduction section of the manuscript:

“To achieve molecular specificity of gas sensing, the sensing mechanism such as the oxygen ionic model is developed. In this model, the oxygen molecules adsorbed can trap the electrons from the ionic oxide to form oxygen ions¹⁶⁻²¹. These oxygen ions are of reducing feature and feasible to react with the oxidizing gas of NO₂, resulting in the releasing of the trapped electron back to the ionic oxide. This carrier-density variation manifests as the sensing response in the oxygen ionic model. Since this indirect surficial doping can vary the resistive signal, the NO₂ concentration can be estimated based on the resistance change. Considering that the surficial oxygen ions are highly active, certain group of the target gases may induce the similar sensing response based on the indirect surficial doping. To improve the specificity, S.-C. Chang proposes the chemical adsorption/desorption model. In this case, the direct doping occurs between the target gas and the sensing block since the adsorbed oxygen is comparatively inert at its dimer state^{21, 22}.”

(2) In the reference section of the manuscript:

“[21] Huang, X., Meng, F., Pi, Z., Xu, W., Liu, J. Gas sensing behavior of a single tin dioxide sensor under dynamic temperature modulation. *Sens. Actuators B Chem.* **99**,

444 (2004).”

1-2. The authors classified three sensing mechanisms in this part, i.e., “oxygen ionic model”, “chemical adsorption/desorption model” and “the model of surface charge transferring”. Actually, these three mechanisms are in essence the same. For practical operation of a gas sensor, interaction with oxygen cannot be excluded, so “direct doping is formed between the target gas and the sensing block” is rare. A sensing process always includes gas adsorption and charge transferring due to gas-solid interaction.

Author reply: Thank the reviewer very much for this comment. I do not agree with that “For practical operation of a gas sensor, interaction with oxygen cannot be excluded”, because oxygen adsorbed exhibits the temperature-dependent chemical activity and becomes comparatively inert at room or low temperature. These oxygen states have been reported to present the temperature-dependent exchanging as follows [Huang, X., Meng, F., Pi, Z., Xu, W., Liu, J. Gas sensing behavior of a single tin dioxide sensor under dynamic temperature modulation. *Sens. Actuators B Chem.* **99**, 444 (2004).]:

where the oxygen dimer state is comparatively inert and thus the direct doping occurs between the target gas and the sensing block. This direct doping is served as the model of surface charge transferring and highly accessible for the sensing block of two-dimensional materials (with Van der Waals surface). Considering that the negligible chemical-reaction occurs for the sensing process on the sample surface, the model of surface charge transferring is also different to the chemical adsorption/desorption model.

To achieve more clear illustration, the related content is updated and pasted below for reading convenience:

(1) In the introduction section of the manuscript:

“To achieve molecular specificity of gas sensing, the sensing mechanism such as the

oxygen ionic model is developed. In this model, the oxygen molecules adsorbed can trap the electrons from the ionic oxide to form oxygen ions¹⁶⁻²¹. These oxygen ions are of reducing feature and feasible to react with the oxidizing gas of NO₂, resulting in the releasing of the trapped electron back to the ionic oxide. This carrier-density variation manifests as the sensing response in the oxygen ionic model. Since this indirect surficial doping can vary the resistive signal, the NO₂ concentration can be estimated based on the resistance change. Considering that the surficial oxygen ions are highly active, certain group of the target gases may induce the similar sensing response based on the indirect surficial doping. To improve the specificity, S.-C. Chang proposes the chemical adsorption/desorption model. In this case, the direct doping occurs between the target gas and the sensing block since the adsorbed oxygen is comparatively inert at its dimer state^{21, 22}. In this case, the p/n-type doping will occur due to the energy difference between molecular hybrid orbital energy level of the gaseous molecular and affinity of the sensing block. Note that the chemical reaction on the block surface induces a long-time and complex sensing response²³. In 2015, J.H. Chen et al. fabricate the two-dimensional material (*i.e.*, phosphorene) into a gas sensor and propose the model of surface charge transferring¹³. By the Van der Waals surface of the two-dimensional material, the sensing response is originated from the charge transferring with the negligible chemical-reaction on the sample surface. This charge transferring model is applied for the gas sensors consisted of low dimensional materials²⁴.”

(2) In the reference section of the manuscript:

“[21] Huang, X., Meng, F., Pi, Z., Xu, W., Liu, J. Gas sensing behavior of a single tin dioxide sensor under dynamic temperature modulation. *Sens. Actuators B Chem.* **99**, 444 (2004).”

2-1. In Fig. 1e and f, the authors present the STM images regarding the adsorption of O₂ molecules on the Se vacancies. Firstly, the sites for Se terminals/vacancies and all the adsorbed O₂ molecules should be clearly indicated on the STM images.

Author reply: Many thanks for this point. We have added the indications in the STM image and pasted the updated as follows for seeing convenience:

In the section “**Two-Dimensional Bi₂O₂Se with Atomically-Flat Surface**” of the manuscript:

“**Fig. 1| Characterizations of Bi₂O₂Se and Its Adsorption Active Sites.** **a** Cross-sectional HAADF-STEM image showing the Bi₂O₂Se grown on the SrTiO₃ (001) substrate (scale bar: 1 nm). **b** The element distributions for the HAADF-STEM image (scale bar: 1 nm). **c** and **d** The AFM images of Bi₂O₂Se surface (**c**; scale bar: 2 μm) and its zoom-in (**d**; scale bar: 400 nm). **e** and **f** The STM images presenting the Bi₂O₂Se surface without (**e**; scale bar: 5 nm) / with adsorbates (**f**; scale bar: 5 nm). The adsorbed O₂ molecules are indicated with red boxes.”

2-2. Secondly, does this structure character have any relationship with the proposed gas sensing mechanism (dipole moment of molecule)?

Author reply: Thank the reviewer very much for this point. The STM characterization is to present the surface morphology of the Bi₂O₂Se, by which the computation about the charge transferring can be conducted with certain surface structure of sensing block.

3. “Note that the resistance in NO₂ is higher than that in vacuum, indicating the p-type

doping/surface scattering originated from the adsorbed NO₂.” The authors describe the increased resistance in NO₂ due to “p-type doping”, do they mean NO₂ molecules are doped onto the surface of Bi₂O₂Se? If so, then the sensor will have a problem to recover, as a dopant in the material is relatively stable.

Author reply: We would like to highly appreciate the reviewer for this comment. We have updated the related content and conducted the computation to reveal that, the NO₂ is served as the adsorbate to induce the charge transferring rather than the dopant in the Bi₂O₂Se. The related is updated and pasted below for viewing convenience:

(1) In section “**Molecularly Specific Detection towards Trace Gases**” of the manuscript:

“Note that this NO₂ doping manifests as the charge transferring from the Bi₂O₂Se to the NO₂ adsorbed, where the NO₂ is served as the adsorbate rather than the dopant for inducing the reconstruction of the Bi₂O₂Se (supplementary information S8).”

(2) In S6 and S8 of the supplementary information:

“Regarding the target gases, Table S1 shows their molecular features such as dipole moment and the highest occupied molecular orbital (HOMO) level. Mulliken charge transferring is extracted for each target gas to clarify the adsorbate doping, and the surface doping can be classified into the n-type and the p-type doping. Quantum mechanical computations are based on the Gaussian 09 computational suite, obeying theory of Coupled Cluster Singles, Doubles, and Perturbative Triples; and the correlation-consistent polarized valence triple-zeta basis set is adopted for a fine balance between computational feasibility and accuracy⁷⁻¹⁰. Additionally, the PBE0 density functional method complemented with the D3(BJ) dispersion correction is employed to conduct all computational analyses. Established by Adamo and Barone in their foundational work on the PBE0 model, this strategy is applied for ensuring accurate and reliable density functional methods without parameter adjustment¹¹⁻¹². Noted for its precise Coulomb-fitting for elements ranging from hydrogen to radon¹³, the def2-SVP basis set is utilized for atoms in the geometry optimizations phase. This basis set choice is subjected to its established accuracy and efficiency. The restricted

optimizations of the crystal structure are conducted and crucial for the investigation. Unrestricted optimizations were performed for specific molecules, including nitrogen dioxide, isopropanol, formaldehyde, methanol, acetone, oxygen, ammonia, ethane, ethanol, acetic acid, formic acid, and acetaldehyde. By the unique properties and behaviors of these molecules, the distinction in methodology shows a thorough and nuanced analysis. Guided by the principles laid out in the aforementioned works, the primary research is focused on the adsorption studies on crystals and small molecules, leveraging the combination of the PBE0 density functional method and the def2-SVP basis set.

Considering that the certain value of $\frac{P_{ads.}-P_0}{P_0}$ ($0 \leq \frac{P_{ads.}-P_0}{P_0} \leq 1$) is accessible for the surface-scattering-dominated sensing mechanism, the NO₂ is characterized with the highest value of $\frac{P_{ads.}-P_0}{P_0}$ (0.07), by which the NO₂ is estimated to form a scattering center to reduce the mobility and enhance the impedance angle at the largest level.

Table S1. The Molecular Features of the Target Gas and Its Adsorbed State

Target molecule	P_0 (Debye)	$P_{ads.}$ (Debye)	$\frac{P_{ads.} - P_0}{P_0}$	HOMO (eV)	Muliken charge transferring
O ₂	0	0	0	-13.16	-0.546
CH ₃ COOH	4.79	4.22	-0.12	-12.27	-0.045
CH ₃ COCH ₃	3.23	2.93	-0.09	-11.18	-0.08
CH ₂ O	2.82	2.49	-0.12	-12.02	-0.085
CH ₃ OH	1.73	1.76	0.02	-12.08	0.21
CH ₃ CH ₂ OH	1.70	1.71	0.01	-12.00	0.215
CHOOH	4.38	3.97	-0.09	-12.70	-0.049
CH ₃ CHOHCH ₃	1.72	1.63	-0.05	-11.86	0.223
NH ₃	1.69	1.55	-0.08	-11.67	0.393
NO ₂	0.45	0.48	0.07	-13.22	-0.259

CH ₃ CH ₂ CHO	3.27	2.60	-0.20	-11.40	-0.099
CH ₃ CH ₃	0	0.32	-	-13.25	0.373

”

“S7. The structure variation for Bi₂O₂Se adsorbed with NO₂

Figure S7 shows the structure variation for Bi₂O₂Se adsorbed with NO₂, where the quantum mechanical computations are conducted based on the Gaussian 09 computational suite. The Bi₂O₂Se remains its structure for NO₂ adsorption and the NO₂ is served as the adsorbate rather than the dopant in the Bi₂O₂Se. Supporting by the charge transferring phenomenon in Table S1, the NO₂ doping manifests as the charge transferring from the Bi₂O₂Se to the NO₂ adsorbed, without the reconstruction of the Bi₂O₂Se.

Fig. S7| The Bi₂O₂Se structure variation for NO₂ adsorption. ”

4. In Fig. 2j, “the resistance is positively correlate to the temperature. The highly-doped feature of the Bi₂O₂Se is therefore clarified”, the authors are suggested to explain clearly the high doping in Bi₂O₂Se.

Author reply: Many thanks to the reviewer for this suggestion. We have added more elaborations and the related reference to explain clearly the highly-doping character of Bi₂O₂Se. Please view the addition as follows for convenience:

(1) In the section “**Fabrication of Schottky-Junction-based Gas Sensor**” of the

manuscript:

“Fig. 2j and Fig. S4c show that the resistance (R) is positively correlated to the temperature (T). This metal-like R - T behavior indicates the highly-doped feature of the $\text{Bi}_2\text{O}_2\text{Se}$, where this highly-doped character is reported to be originated from the self-modulation doping effect²⁷. The highly-doped feature of the $\text{Bi}_2\text{O}_2\text{Se}$ is further supported by the negative relation between the mobility of $\text{Bi}_2\text{O}_2\text{Se}$ and temperature (Fig. 2k), since the temperature increase enhances the carrier scattering effect.”

(2) In the reference section of the manuscript:

“[27] Fu, H., Wu, J., Peng, H., Yan, B. Self-modulation doping effect in the high-mobility layered semiconductor $\text{Bi}_2\text{O}_2\text{Se}$. *Phys. Rev. B* **97**, 241203 (R) (2018).”

5. Fig.3, “The resistance and reactance value increase for the p-type doping of adsorbates.” This is not the fact, as one can see in Fig. 3c and d that reactance decreases upon exposing to methanol/methanal.

Author reply: Thank the reviewer very much for this typo issue, and the related content is corrected and pasted below for viewing convenience:

In the section “**Molecularly Specific Detection towards Trace Gases**” of the manuscript:

“Related to the methanal and acetone, the absolute values of resistance and reactance increase for the p-type doping of adsorbates (see supplementary information Table S1). For the methanol, its surface scattering effect suppresses the n-type doping and induces the enhanced resistance. Regarding the impedance angle, it is dominated by the surface scattering and its absolute value increases for more adsorbates (*e.g.*, methanol, methanal, and acetone).”

6. Figs. 3 and 4, why the resistance of $\text{Bi}_2\text{O}_2\text{Se}$ increases upon exposing to typical reducing gases such as methanol, methanal, isopropanol and acetone. These reducing gases behaves just like oxidizing NO_2 .

Author reply: Many thanks for the reviewer for this comment. We have conducted

the computation to reveal the charge transferring between the target gas and the Bi₂O₂Se. It is found that, just like the NO₂, p-type doping occurs for the methanal and the acetone adsorbed, indicating these two gases serve as the oxidizing gas. Regarding the resistance response to isopropanol/methanol, it is attributed to the scattering effect formed by the adsorbate on the Bi₂O₂Se surface, which suppresses their n-type doping effect. The related content is added into the SI and pasted herein for viewing convenience:

(1) In the section “**Molecularly Specific Detection towards Trace Gases**” of the manuscript:

“Related to the methanal and acetone, the absolute values of resistance and reactance increase for the p-type doping of adsorbates (see supplementary information Table S1). For the methanol, its surface scattering effect suppresses the n-type doping and increases the resistance. Regarding the impedance angle, it is dominated by the surface scattering and its absolute value increases for more adsorbates (*e.g.*, methanol, methanal, and acetone).”

(2) In S6 of the supplementary information:

“Regarding the target gases, Table S1 shows their molecular features such as dipole moment and the highest occupied molecular orbital (HOMO) level. Mulliken charge transferring is extracted for each target gas to clarify the adsorbate doping, and the surface doping can be classified into the n-type and the p-type doping. Quantum mechanical computations are based on the Gaussian 09 computational suite, obeying theory of Coupled Cluster Singles, Doubles, and Perturbative Triples; and the correlation-consistent polarized valence triple-zeta basis set is adopted for a fine balance between computational feasibility and accuracy⁷⁻¹⁰. Additionally, the PBE0 density functional method complemented with the D3(BJ) dispersion correction is employed to conduct all computational analyses. Established by Adamo and Barone in their foundational work on the PBE0 model, this strategy is applied for ensuring accurate and reliable density functional methods without parameter adjustment¹¹⁻¹². Noted for its precise Coulomb-fitting for elements ranging from hydrogen to radon¹³,

the def2-SVP basis set is utilized for atoms in the geometry optimizations phase. This basis set choice is subjected to its established accuracy and efficiency. The restricted optimizations of the crystal structure are conducted and crucial for the investigation. Unrestricted optimizations were performed for specific molecules, including nitrogen dioxide, isopropanol, formaldehyde, methanol, acetone, oxygen, ammonia, ethane, ethanol, acetic acid, formic acid, and acetaldehyde. By the unique properties and behaviors of these molecules, the distinction in methodology shows a thorough and nuanced analysis. Guided by the principles laid out in the aforementioned works, the primary research is focused on the adsorption studies on crystals and small molecules, leveraging the combination of the PBE0 density functional method and the def2-SVP basis set.

Considering that the certain value of $\frac{P_{ads.}-P_0}{P_0}$ ($0 \leq \frac{P_{ads.}-P_0}{P_0} \leq 1$) is accessible for the surface-scattering-dominated sensing mechanism, the NO₂ is characterized with the highest value of $\frac{P_{ads.}-P_0}{P_0}$ (0.07), by which the NO₂ is estimated to form a scattering center to reduce the mobility and enhance the impedance angle at the largest level.

Table S1. The Molecular Features of the Target Gas and Its Adsorbed State

Target molecule	P_0 (Debye)	$P_{ads.}$ (Debye)	$\frac{P_{ads.} - P_0}{P_0}$	HOMO (eV)	Muliken charge transferring
O ₂	0	0	0	-13.16	-0.546
CH ₃ COOH	4.79	4.22	-0.12	-12.27	-0.045
CH ₃ COCH ₃	3.23	2.93	-0.09	-11.18	-0.08
CH ₂ O	2.82	2.49	-0.12	-12.02	-0.085
CH ₃ OH	1.73	1.76	0.02	-12.08	0.21
CH ₃ CH ₂ OH	1.70	1.71	0.01	-12.00	0.215
CHOOH	4.38	3.97	-0.09	-12.70	-0.049
CH ₃ CHOHCH ₃	1.72	1.63	-0.05	-11.86	0.223
NH ₃	1.69	1.55	-0.08	-11.67	0.393

NO ₂	0.45	0.48	0.07	-13.22	-0.259
CH ₃ CH ₂ CHO	3.27	2.60	-0.20	-11.40	-0.099
CH ₃ CH ₃	0	0.32	-	-13.25	0.373

”

7. Poor description in the manuscript, e.g., “the sensitivity of the resistive”, “As the concentration of the mixed increases from 33% to 50 %, the sensitivities of the multiple signals (involving R) reduce for the NO₂ detection”.

Author reply: We would like to highly appreciate the reviewer for this comment and add more descriptions accordingly. Please see the pasted below for the updated:

In the section “**Molecularly Specific Detection towards Trace Gases**” of the manuscript:

“The molecular specificity of the impedance angle is further demonstrated when the target gas NO₂ is mixed with the other gas. Firstly, the trace NO₂ (430 ppt) is mixed with the low concentration O₂ ($C(\text{NO}_2)/C(\text{O}_2) = 1 \times 10^{-4}$) and served as target gas in Fig. 4b. Compared to the responses to the NO₂ target gas, the resistive sensitivity to the mixture almost remains (10.5 %·ppb⁻¹), while the reactance and the impedance-angle sensitivity are specific to the mixed gas (the reactance: 23.3 %·ppb⁻¹; and the impedance: 12.1 %·ppb⁻¹). Considering the fact that the responses of the reactance and the impedance angle are sensitive to the mixed gas, the concentration of the mixed (e.g., methanol or methanal) is increased up to 33% and 50 % for the study on the molecular specificity. Fig. 4b presents that, in comparison with response to the NO₂, the sensitivities of the multiple signals reduce for the gas-mixture detection, indicating the existence of the mixed gases. In the case of the NO₂ mixed with 50 % of methanal, the sensitivities are respectively decreased by 75 %, 30 %, and 80 % for those of the resistance, the reactance, and the impedance angle. In contrast to the sensing signals of the resistance and the reactance, the impedance angle is more sensitive to each gas in the mixture and featured with molecular specificity.”

8. In S6, “(1) the surface scattering usually interferes the resistive signal and degrades the sensitivity;” Take the response of Bi₂O₂Se toward NO₂ for an example, on the one hand, “the adsorbed NO₂ decreases the carrier density of the Bi₂O₂Se (n)”, on the other hand, the adsorbed NO₂ molecules form as the scattering centers and reduce the carrier mobility (μ) of Bi₂O₂Se. As both n and μ decrease, a even higher resistance is obtained for Bi₂O₂Se, and a larger response to NO₂ is achieved. Therefore, in this case, the surface scattering does not interfere the resistive signal, but reinforces it.

Author reply: Thank the reviewer very much for this comment. I agree with that the surface scattering reinforce the resistive signal in the NO₂ detection. However, by this surface scattering, the resistive signal cannot show the specificity to NO₂ by the differential doping to the Bi₂O₂Se. Thus, in the case related to the specificity optimization of the resistive signal, the surface scattering is the interference.

9-1. The authors claimed that they achieved “Molecularly Specific Detection of NO₂” by measuring the impedance angle, which is dependent on the dipole moment of the target molecule and featured with molecular specificity. However, as seen in Fig. 4, all gases can give a signal for the parameter of “impedance angle”, which can seriously interfere the detection of NO₂.

Author reply: Thank the reviewer very much for the comments. The impedance angle is related to the change in dipole moment of the target molecule adsorbed, which is differential to each molecule. More details are presented in Table S1, which is also pasted following for viewing convenience. In Fig. 4, the sensitivity of impedance angle is differential to each gas and featured with the molecularly specificity rather than the interference. Please viewing the Table S1 and Fig. 4 as follows:

(1) In S6 of the supplementary information:

“Table S1. The Molecular Features of the Target Gas and Its Adsorbed State

Target molecule	P_0 (Debye)	$P_{ads.}$ (Debye)	$\frac{P_{ads.} - P_0}{P_0}$	HOMO (eV)	Muliken charge transferring
-----------------	------------------	-----------------------	------------------------------	--------------	--------------------------------

O ₂	0	0	0	-13.16	-0.546
CH ₃ COOH	4.79	4.22	-0.12	-12.27	-0.045
CH ₃ COCH ₃	3.23	2.93	-0.09	-11.18	-0.08
CH ₂ O	2.82	2.49	-0.12	-12.02	-0.085
CH ₃ OH	1.73	1.76	0.02	-12.08	0.21
CH ₃ CH ₂ OH	1.70	1.71	0.01	-12.00	0.215
CHOOH	4.38	3.97	-0.09	-12.70	-0.049
CH ₃ CHOHCH ₃	1.72	1.63	-0.05	-11.86	0.223
NH ₃	1.69	1.55	-0.08	-11.67	0.393
NO ₂	0.45	0.48	0.07	-13.22	-0.259
CH ₃ CH ₂ CHO	3.27	2.60	-0.20	-11.40	-0.099
CH ₃ CH ₃	0	0.32	-	-13.25	0.373

”

(2) In section “**Molecularly Specific Detection towards Trace Gases**” of the manuscript:

“

Fig. 4]. Sensing Responses to Different Gases. a and b The responses (estimated by signals of R , X , and Θ) to different gases (**a**; isopropanol (Iso.), acetone, NO_2) and NO_2 mixture (**b**; NO_2/O_2 , $\text{NO}_2/\text{CH}_3\text{OH}$, $\text{NO}_2/\text{CH}_2\text{O}$).”

9-2. The method to reach “Molecularly Specific Detection of NO_2 ” is by PCA, an approach based on statistical analysis. This means the method proposed in this work is not molecularly specific, but a result of statistical analysis. A typical example for

molecularly specific detection is non-dispersive infrared gas sensors, which analyze the concentration of target gases based on their characteristic infrared absorption. The detection of a target gas cannot be interfered by other ones.

Author reply: Many thanks for this comment. This investigation applies the PCA analysis to enhance the variance of the sensing signals (especially for the impedance angle), aiming for presenting the molecular characters distinctly. Based on the fitted data of the impedance angle (*i.e.*, slope and intercept originated from the linear fitting), The PCA analysis is conducted to show that, the projections on the principle component 1 and 2 can be differential to indicate the molecular characteristics.

Importantly, to enhance the variance (α) in PCA analysis, the impedance angle is analyzed with the other sensing signals (*i.e.*, the resistive and the reactance). In this case, the variance (α) originated from the multiple sensing signals is up to α^6 ($\alpha > 1$), which is 4 orders of magnitude larger than that of the traditional resistance. And its three-dimensional component plot shows the none-overlaid projections on the principle component 1-3 are differential for each gas (Fig. 5a). In contrast, Fig. 5b presents the principle component analysis based on the fitted data of the resistance (*i.e.*, slope and intercept originated from the linear fitting), where the projections are highly overlaid in the two-dimensional plot.

Reviewer #3 (Remarks to the Author):

The manuscript describes the development of a sensor device for determining ultra-low concentrations of nitrogen dioxide. The authors propose to use the impedance angle, which exhibits maximum changes when exposed to gas, as a sensor signal. This experimental fact is explained based on the model of surface scattering of charge carriers during adsorption of NO₂ molecules. In order to ensure the specificity of the gas detection, signals were processed using the of the principal component analysis. The practically important results are related to the choice of two-dimensional bismuth oxyselenide as a sensor material and the appropriate electrodes for the formation of the Schottky barrier. The introduction confirms the importance of this

research. A comprehensive characterization of the synthesized material was carried out, all the conclusions were explained.

The authors should improve the article, as the manuscript contains quite a lot of grammatical errors. For example, there is a repeated error in the word "impedance".

Author reply: Many thanks for the reviewer for this question, and we have corrected accordingly.

It is also necessary to correct some mistakes in the presented figures: Figure S1(c) is uninformative; Figure S2(b) does not show the peak of Se at 53 eV.

Author reply: Thank the reviewer very much for this point. We have added the in-plane view of the EBSD mapping into the Figure S1(c). Additionally, the notation for the peak of Se at 53 eV is added into Figure S2 (b). The updated content is pasted below for viewing convenience:

In S1 and S2 of the supplementary information:

“S1. The Epitaxial Growth of Bi₂O₂Se on the SrTiO₃ Substrate

Figure S1a shows the X-ray diffraction (XRD) pattern of the Bi₂O₂Se grown on the SrTiO₃ (001) substrate. Towards out of plane, the SrTiO₃ substrate and the Bi₂O₂Se present lattice planes of (001). The Bi₂O₂Se shows lattice planes of (002), (004), (006), (008), and (0010). Fig. S1b is the scanning electron microscopy (SEM) image of the Bi₂O₂Se sheet, and it is etched to show the rectangle shape. This sample is applied for electron back scatter diffraction (EBSD) characterization. From the in-plane and the out-of-plane view, (001) lattice plane is observed for Bi₂O₂Se and SrTiO₃ (Fig. S1c). The Bi₂O₂Se is demonstrated to be epitaxially grown along the (001) crystal axis on the SrTiO₃ (001) substrate.

Fig. S1| a XRD pattern of Bi₂O₂Se on the SrTiO₃ (001) substrate. **b** and **c** SEM image of the Bi₂O₂Se (**b**; scale bar: 10 μm) and its corresponding EBSD pattern from the in-plane and the out-of-plane view (**c**; scale bar: 10 μm). ”

“

Fig. S2| a and b The UPS (a) and the XPS pattern (b) of the $\text{Bi}_2\text{O}_2\text{Se}$.”

Since the exhaled air always contains a large amount of water vapors, it is interesting to investigate their effect on the trace NO_2 detection characteristics.

Author reply: Many thanks to the reviewer for this comment. Limited by the experimental setup for trace NO_2 detection characteristics, it is highly difficult to investigate the effect of water vapors. Thus, we have reduced the description about potential application into the disease marker detection. The main updated is shown in the section of introduction, which is pasted below for reading convenience:

“Trace NO_2 is featured with high chemical activity and its concertation variation is usually correlated with the issues among the production and life¹⁻⁴. Nowadays the trace NO_2 detection mainly relies on gas chromatography-mass spectrometer technology, adsorption photometry, and gas sensing⁵⁻¹². The former two are of ppt detection limit but still show two issues: (i) the related device cannot be accessible for real-time monitoring and portability; (ii) the error originated from gas-concentration characterization is nontrivial due to the long-time gas collection^{5,6}. In comparison, the gas sensor is integratable and featured with rapid response. Now its sensitivity is well developed and appropriate for trace gas detection. The further development of gas sensors is limited by lack of highly specific sensing mechanism⁷⁻¹⁵. Molecular character shall be outputted if certain kinds of the **circumambient** gases induced the sensing response similar to that of NO_2 .”

Reviewers' Comments:

Reviewer #1:

Remarks to the Author:

The authors argued that impedance serves as an indicator for specificity detection in the interactions between adsorbed gases and Bi₂O₂Se. As a basis for this, it was suggested through calculation that the larger the dipole moment change of the gas after adsorption, the greater the contribution to the mobility change by acting as the scattering center of the surface. These revisions by the authors provide additional evidence for the manuscript's core content, thereby enhancing its persuasiveness. Furthermore, the manuscript has been revised with easy-to-understand expressions suitable for the multidisciplinary audience of Nature Communications, ensuring comprehensibility across various fields.

However, in parts not directly related to the author's central thesis, questions arise regarding the formulas and methodologies used in the Schottky Barrier Height (SBH) calculations as follows:

Q1. The authors' claim regarding the change in the Fermi level (E_f) as inferred through the Arrhenius equation raises questions:

1) Appropriateness of the SBH equation.

The typical bandgap of Bi₂O₂Se is known to be 0.8 eV.[1] If the SBH for the extracted electron exceeds 2.7 eV, the band diagram would not resemble Figure 2d, g. Instead, the W_f of Au would be positioned significantly below Bi₂O₂Se's Valence Band Maximum. This scenario appears to be quite distant from typical junction behavior. This necessitates a verification of the appropriateness of the Arrhenius equation used for SBH, along with the addition of relevant references to support its application.

2) Doping dependence of the SBH.

Accurate SBH extraction between electrode-2D materials requires identifying a section where SBH changes linearly with the change in Gate Voltage (VG), extracting SBH at the VG=VFB (Flat Band Voltage) starting point. If the doping effect by gas changes E_f , leading to a shift in the VFB point, it could indicate a change in E_f due to gas adsorption.

However, as the author argued, the junction between Au-Bi₂O₂Se forms a Schottky junction (Bi₂O₂Se W_f 5.2 eV). Therefore, the change in SBH cannot be considered as an E_f change because it is not a linear range of VG-SBH when VG=0. (Refer to Figure 4 of the attached by Allain et al., 2015).[2]

[1] Xu, S. et al. Exploiting Two-Dimensional Bi₂O₂Se for Trace Oxygen Detection. *Angew. Chem. Int. Ed.* 59, 17938–17943 (2020).

[2] Allain, A., Kang, J., Banerjee, K. & Kis, A. Electrical contacts to two-dimensional semiconductors. *Nat. Mater.* 14, 1195–1205 (2015).

Q2. Certain sentence expressions need revision, particularly in the "Molecularly Specific Detection towards Trace Gases" section, "For methanol, its surface scattering effect suppresses the n-type doping and increases the resistance."

- Recommendations:

R1. Review the use of the term "doping." Since "doping" is defined as "The action of adding a small amount of foreign atoms to form a solid solution in the lattice of a non-metallic catalyst,"[3] there might be misunderstandings regarding internal penetration of gas molecules into the channel. If finding an appropriate word is challenging, it might be beneficial to precisely explain the use of the term "doping" in this manuscript.

[3] 'doping' in IUPAC Compendium of Chemical Terminology, 3rd ed. International Union of Pure and Applied Chemistry; 2006. Online version 3.0.1, 2019.

<https://doi.org/10.1351/goldbook.D01834>

R2. A thorough review of the narrative consistency throughout the document is recommended. Given the changes in narrative during the revision process, ensuring consistency across the manuscript can further improve its readability.

Reviewer #2:

Remarks to the Author:

The authors have addressed my concerns towards the previous manuscript. This revised manuscript is acceptable for publication on Nature Communications.

Reviewer #3:

Remarks to the Author:

The authors carefully revised the manuscript, making all necessary changes based on the reviewers' comments and suggestions.

Point-by-point response to the reviewer comments

Many thanks to the reviewers for their comments. These comments are highly important and have been replied point-by-point, by which the manuscript is remarkably strengthened. Our reply and the updated contents are presented as follows. For viewing convenience, the reviewer comments are highlighted in violet; the author reply and the updated are marked in black and blue, respectively.

REVIEWER COMMENTS

Reviewer #1 (Remarks to the Author):

The authors argued that impedance serves as an indicator for specificity detection in the interactions between adsorbed gases and Bi₂O₂Se. As a basis for this, it was suggested through calculation that the larger the dipole moment change of the gas after adsorption, the greater the contribution to the mobility change by acting as the scattering center of the surface. These revisions by the authors provide additional evidence for the manuscript's core content, thereby enhancing its persuasiveness. Furthermore, the manuscript has been revised with easy-to-understand expressions suitable for the multidisciplinary audience of Nature Communications, ensuring comprehensibility across various fields.

However, in parts not directly related to the author's central thesis, questions arise regarding the formulas and methodologies used in the Schottky Barrier Height (SBH) calculations as follows:

Author reply: We would like to highly appreciate the reviewer for the important comments. These comments have been responded point-by-point and pasted below for viewing convenience:

Q1. The authors' claim regarding the change in the Fermi level (E_f) as inferred through the Arrhenius equation raises questions:

1) Appropriateness of the SBH equation.

1-1) The typical bandgap of Bi₂O₂Se is known to be 0.8 eV.^[1] If the SBH for the extracted electron exceeds 2.7 eV, the band diagram would not resemble Figure 2d, g.

Instead, the W_f of Au would be positioned significantly below $\text{Bi}_2\text{O}_2\text{Se}$'s Valence Band Maximum.

Author reply: Thank the reviewer very much for this point. We have revised the band diagram in Figure 2d and g, according to the reviewer's important comment. Additionally, this investigation is focused on the study on the forward-biased contact and thus the relevant band diagram is presented with indications in the caption. The updated Figure 2 is pasted below for viewing convenience:

In section "Fabrication of Schottky-Junction-based Gas Sensor" of the manuscript:

“

Fig. 2| Adsorbate-Induced Variation of the Schottky-Junction Barrier. **a** The photograph of the Bi₂O₂Se gas sensor (scale bar: 12 μm). **b** AFM image showing topology of the Bi₂O₂Se-Au contact (scale bar: 3 μm). **c** The KPFM pattern of Bi₂O₂Se-Au contact (scale bar: 3 μm). **d** The energy band profile of the forward-biased contact in vacuum (vac.) (ϕ_{Au} : work function of Au; E_F : Fermi level of Bi₂O₂Se; E_c : conduction band minimum; E_v : valence band maximum). **e** and **f** The temperature-dependent I - V plot of the device in vacuum and its linear fitting (**f**, $\ln(I/T^2)$ and T^{-1}). **g** The energy band profile of the forward-biased contact in NO₂ (700 ppt). **h** The I - V relation as the function of temperature in NO₂ (700 ppt). **i** The linear fitting between $\ln(I/T^2)$ and T^{-1} for I - V curve in **h**. **j** and **k** The temperature-dependent resistance (**j**) and mobility (**k**) in Ar environment. **l** The linear fitting for extraction of ionization energy (the measurement conducted in vacuum).”

1-2) This scenario appears to be quite distant from typical junction behavior. This necessitates a verification of the appropriateness of the Arrhenius equation used for SBH, along with the addition of relevant references to support its application.

Author reply: Many thanks to the reviewer for this comment. This investigation studies the junction behavior for the forward-biased contact, rather than the reversed-biased contact. In the forward-biased contact, the net electron (injected from the semiconductor to the metal electrode) shows exponential relation with SBH, obeying the thermionic emission theory. The relevant Arrhenius equation is verified by Equation 59 at the chapter 3 of the *Physics of Semiconductor Devices* (Fourth Edition). In the manuscript (including the supplementary information), the related contents are updated with indication that the contact is forward-biased, along with the reference aforementioned to verify the appropriateness of the Arrhenius equation. Please see the updated as follows for convenience:

(1) In section “**Fabrication of Schottky-Junction-based Gas Sensor**” of the manuscript:

“The Bi₂O₂Se is bonded to Au electrode (Figure 2a) to form with the Schottky junction under the forward bias, where the net electron is injected into the Au and

obeys the thermionic emission theory²⁷.”

(2) In section “**Fabrication of Schottky-Junction-based Gas Sensor**” of the manuscript:

“The variation of SBH is estimated by fitting $\ln(I/T^2)$ and T^{-1} , according to the thermionic emission theory. In details, the thermionic emission theory describes the correlation between the current I and SBH (ΔE) as follows²⁷:

$$I = SA^*T^2 \exp\left(\frac{-\Delta E}{k_B T}\right) \left[\exp\left(\frac{qV_{ds}}{k_B T}\right) - 1\right] \quad (1)$$

where S , A^* , q , and k_B are respectively contact area, Richard coefficient, charge quantity, and Boltzmann constant. Note that when the applied voltage V_{ds} is large enough and the equation (1) can be simplified as equation (2) below:

$$I = SA^*T^2 \exp\left(\frac{qV_{ds} - \Delta E}{k_B T}\right) \quad (2)$$

where the slope extracted by the linearly fitting between $\ln(I/T^2)$ and T^{-1} is served as the value of $(qV_{ds} - \Delta E)/k_B$, and the SBH value ΔE is obtained with the certain applied voltage V_{ds} . In NO_2 , the ΔE (NO_2) is then estimated to be 2.795 eV (V_{ds} : 3 V) which is lower than that in vacuum (ΔE (Vac.): 2.846 eV). It is indicated that the adsorbed NO_2 decreases the carrier density of the $\text{Bi}_2\text{O}_2\text{Se}$ (n_0) and the SBH reduces by 0.051 eV.”

(3) In section “**Fabrication of Schottky-Junction-based Gas Sensor**” of the manuscript:

“

Fig. 2 | Adsorbate-Induced Variation of the Schottky-Junction Barrier. **a** The photograph of the $\text{Bi}_2\text{O}_2\text{Se}$ gas sensor (scale bar: $12\ \mu\text{m}$). **b** AFM image showing topology of the $\text{Bi}_2\text{O}_2\text{Se}$ -Au contact (scale bar: $3\ \mu\text{m}$). **c** The KPFM pattern of $\text{Bi}_2\text{O}_2\text{Se}$ -Au contact (scale bar: $3\ \mu\text{m}$). **d** The energy band profile of the forward-biased contact in vacuum (vac.) (ϕ_{Au} : work function of Au; E_{F} : Fermi level of $\text{Bi}_2\text{O}_2\text{Se}$; E_{c} : conduction band minimum; E_{v} : valence band maximum). **e** and **f** The temperature-dependent I - V plot of the device in vacuum and its linear fitting (**f**, $\ln(I/T^2)$ and T^{-1}). **g** The energy band profile of the forward-biased contact in NO_2 (700 ppt). **h** The I - V relation as the function of temperature in NO_2 (700 ppt). **i** The linear fitting

between $\ln(I/T^2)$ and T^{-1} for I - V curve in **h**, **j** and **k** The temperature-dependent resistance (**j**) and mobility (**k**) in Ar environment. **l** The linear fitting for extraction of ionization energy (the measurement conducted in vacuum).”

(4) In the reference section of the manuscript:

“[27] Sze, S.M., Li, Y., Ng, K.K. *Physics of Semiconductor Devices* (Wiley, Fourth Edition).”

(5) In the S11 section of the supplementary information:

“To extract the Schottky barrier height, a linearly fitting between the $\ln(I/T^2)$ and T^{-1} is performed on the forward-biased contact and the slope is the value of $(qV_{ds} - \Delta E)/k_B$, according to the thermionic emission theory²².”

(6) In the reference section of the supplementary information:

“[22] Sze, S.M., Li, Y., Ng, K.K. *Physics of Semiconductor Devices* (Wiley, Fourth Edition).”

2) Doping dependance of the SBH.

2-1) Accurate SBH extraction between electrode-2D materials requires identifying a section where SBH changes linearly with the change in Gate Voltage (V_G), extracting SBH at the $V_G = V_{FB}$ (Flat Band Voltage) starting point. If the doping effect by gas changes E_f , leading to a shift in the V_{FB} point, it could indicate a change in E_f due to gas adsorption.

Author reply: Many thanks to the reviewer for this comment. In the Figure 4 of the work reported by A. Allain et al. [*Nat. Mater.* 14, 1195 (2015)], the method for extracting the SBH is accessible for the reverse-biased contact, while the strategy to extract the SBH (ΔE) in our investigation is focused on the forward-biased contact. For the forward-biased contact, the net electron is injected from the semiconductor to the metal electrode, and its injection is limited by the built-in potential formed among the depletion region of the semiconductor. Obeying the thermionic emission theory

[Sze, S.M., Li, Y., Ng, K.K. *Physics of Semiconductor Devices* (Wiley, Fourth Edition).], the current intensity I is correlated with the forward bias V_{ds} as the formulas below:

$$I = SA^*T^2 \exp\left(\frac{-\Delta E}{k_B T}\right) \left[\exp\left(\frac{qV_{ds}}{k_B T}\right) - 1\right] \quad (1)$$

where S , A^* , k_B , and q are respectively contact area, Richard coefficient, Boltzmann constant, and charge quantity. When the applied voltage V_{ds} is large enough, the equation (1) can be simplified as equation (2) as follows:

$$I = SA^*T^2 \exp\left(\frac{qV_{ds} - \Delta E}{k_B T}\right) \quad (2)$$

where the value of $(qV_{ds} - \Delta E)/k_B$ is equaled to the slope extracted by the linearly fitting between $\ln(I/T^2)$ and T^{-1} , and the SBH ΔE is acquired with the certain forward voltage V_{ds} .

We have indicated that the contact is forward biased in the manuscript, along with the reference for verification. The updated contents are pasted below for reading convenience:

(1) In section “**Fabrication of Schottky-Junction-based Gas Sensor**” of the manuscript:

“The Bi₂O₂Se is bonded to Au electrode (Figure 2a) to form with the Schottky junction under the forward bias, where the net electron is injected into the Au and obeys the thermionic emission theory²⁷.”

(2) In section “**Fabrication of Schottky-Junction-based Gas Sensor**” of the manuscript:

“The variation of SBH is estimated by fitting $\ln(I/T^2)$ and T^{-1} , according to the thermionic emission theory. In details, the thermionic emission theory describes the correlation between the current I and SBH (ΔE) as follows²⁷:

$$I = SA^*T^2 \exp\left(\frac{-\Delta E}{k_B T}\right) \left[\exp\left(\frac{qV_{ds}}{k_B T}\right) - 1\right] \quad (1)$$

where S , A^* , q , and k_B are respectively contact area, Richard coefficient, charge quantity, and Boltzmann constant. Note that when the applied voltage V_{ds} is large enough and the equation (1) can be simplified as equation (2) below:

$$I = SA^*T^2 \exp\left(\frac{qV_{ds} - \Delta E}{k_B T}\right) \quad (2)$$

where the slope extracted by the linearly fitting between $\ln(I/T^2)$ and T^{-1} is served as the value of $(qV_{ds} - \Delta E)/k_B$, and the SBH value ΔE is obtained with the certain applied voltage V_{ds} . In NO_2 , the ΔE (NO_2) is then estimated to be 2.795 eV (V_{ds} : 3 V) which is lower than that in vacuum (ΔE (Vac.): 2.846 eV). It is indicated that the adsorbed NO_2 decreases the carrier density of the $\text{Bi}_2\text{O}_2\text{Se}$ (n_0) and the SBH reduces by 0.051 eV.”

(3) In section “**Fabrication of Schottky-Junction-based Gas Sensor**” of the manuscript:

“

Fig. 2 | Adsorbate-Induced Variation of the Schottky-Junction Barrier. **a** The photograph of the Bi₂O₂Se gas sensor (scale bar: 12 μm). **b** AFM image showing topology of the Bi₂O₂Se-Au contact (scale bar: 3 μm). **c** The KPFM pattern of Bi₂O₂Se-Au contact (scale bar: 3 μm). **d** The energy band profile of the forward-biased contact in vacuum (vac.) (ϕ_{Au} : work function of Au; E_{F} : Fermi level of Bi₂O₂Se; E_{c} : conduction band minimum; E_{v} : valence band maximum). **e** and **f** The temperature-dependent I - V plot of the device in vacuum and its linear fitting (**f**, $\ln(I/T^2)$ and T^{-1}). **g** The energy band profile of the forward-biased contact in NO₂ (700 ppt). **h** The I - V relation as the function of temperature in NO₂ (700 ppt). **i** The linear fitting

between $\ln(I/T^2)$ and T^{-1} for I - V curve in **h**. **j** and **k** The temperature-dependent resistance (**j**) and mobility (**k**) in Ar environment. **l** The linear fitting for extraction of ionization energy (the measurement conducted in vacuum).”

(4) In the reference section of the manuscript:

“[27] Sze, S.M., Li, Y., Ng, K.K. *Physics of Semiconductor Devices* (Wiley, Fourth Edition).”

2-2) However, as the author argued, the junction between Au-Bi₂O₂Se forms a Schottky junction (Bi₂O₂Se W_f 5.2 eV). Therefore, the change in SBH cannot be considered as an E_f change because it is not a linear range of V_G -SBH when $V_G = 0$. (Refer to Figure 4 of the attached by Allain et al., 2015).[2]

[1] Xu, S. et al. Exploiting Two-Dimensional Bi₂O₂Se for Trace Oxygen Detection. *Angew. Chem. Int. Ed.* 59, 17938-17943 (2020).

[2] Allain, A., Kang, J., Banerjee, K. & Kis, A. Electrical contacts to two-dimensional semiconductors. *Nat. Mater.* 14, 1195-1205 (2015).

Author reply: Thank the reviewer very much for this comment. In our investigation, the strategy to extract the SBH is focused on the forward-biased contact, rather than the reverse-biased contact in the Figure 4 of the work conducted by A. Allain et al. [*Nat. Mater.* 14, 1195 (2015)]. Under the forward bias, the SBH can be acquired by the thermionic emission theory [Sze, S.M., Li, Y., Ng, K.K. *Physics of Semiconductor Devices* (Wiley, Fourth Edition).], **where the nonlinear effects such as the image-force lowering and the tunnelling current are trivial** (Figure 13 and 3.3.4 section in the Chapter 3 of the *Physics of Semiconductor Devices*). The SBH therefore can be extracted from the linearly fitting between $\ln(I/T^2)$ and T^{-1} , according to the equation 2. In this mean, the variation of SBH is estimated for the p-doping of the NO₂, **where the Fermi level variation of the Bi₂O₂Se is not indicated**. The related contents are updated in the manuscript and pasted following for viewing convenience:

(1) In section “**Fabrication of Schottky-Junction-based Gas Sensor**” of the manuscript:

“The variation of SBH is estimated by fitting $\ln(I/T^2)$ and T^{-1} , according to the thermionic emission theory. In details, the thermionic emission theory describes the correlation between the current I and SBH (ΔE) as follows²⁷:

$$I = SA^*T^2 \exp\left(\frac{-\Delta E}{k_B T}\right) \left[\exp\left(\frac{qV_{ds}}{k_B T}\right) - 1\right] \quad (1)$$

where S , A^* , q , and k_B are respectively contact area, Richard coefficient, charge quantity, and Boltzmann constant. Note that when the applied voltage V_{ds} is large enough and the equation (1) can be simplified as equation (2) below:

$$I = SA^*T^2 \exp\left(\frac{qV_{ds} - \Delta E}{k_B T}\right) \quad (2)$$

where the slope extracted by the linearly fitting between $\ln(I/T^2)$ and T^{-1} is served as the value of $(qV_{ds} - \Delta E)/k_B$, and the SBH value ΔE is obtained with the certain applied voltage V_{ds} . In NO_2 , the ΔE (NO_2) is then estimated to be 2.795 eV (V_{ds} : 3 V) which is lower than that in vacuum (ΔE (Vac.): 2.846 eV). It is indicated that the adsorbed NO_2 decreases the carrier density of the $\text{Bi}_2\text{O}_2\text{Se}$ (n_0) and the SBH reduces by 0.051 eV.”

(2) In the reference section of the manuscript:

“[27] Sze, S.M., Li, Y., Ng, K.K. *Physics of Semiconductor Devices* (Wiley, Fourth Edition).”

Q2. Certain sentence expressions need revision, particularly in the "Molecularly Specific Detection towards Trace Gases" section, "For methanol, its surface scattering effect suppresses the n-type doping and increases the resistance."

Author reply: Many thanks to the reviewer for this point. The related content is revised and pasted following for checking convenience:

In section “**Molecularly Specific Detection towards Trace Gases**” of the manuscript:

“For the methanol, its surface scattering effect suppresses the n-type doping, by which the sensor resistance increases.”

- Recommendations:

R1. Review the use of the term "doping." Since "doping" is defined as "The action of adding a small amount of foreign atoms to form a solid solution in the lattice of a non-metallic catalyst,"[3] there might be misunderstandings regarding internal penetration of gas molecules into the channel. If finding an appropriate word is challenging, it might be beneficial to precisely explain the use of the term "doping" in this manuscript.

[3] 'doping' in IUPAC Compendium of Chemical Terminology, 3rd ed. International Union of Pure and Applied Chemistry; 2006. Online version 3.0.1, 2019. <https://doi.org/10.1351/goldbook.D01834>

Author reply: Thank the reviewer very much for this suggestion. We have added the elaboration to explain the use of the term "doping" in this manuscript. The addition is pasted as follows for reading convenience:

In the introduction section of the manuscript:

“This carrier-density variation manifests as the sensing response in the oxygen ionic model. Since this indirect surficial doping can vary the resistive signal, the NO₂ concentration can be estimated based on the resistance change. Note that the surficial doping for the sensing process manifests by the electron exchanging between the adsorbate and sensing block, rather than the internal penetration of the gas molecules into the channel. Considering that the surficial oxygen ions are highly active, certain group of the target gases may induce the similar sensing response based on the indirect surficial doping.”

R2. A thorough review of the narrative consistency throughout the document is recommended. Given the changes in narrative during the revision process, ensuring consistency across the manuscript can further improve its readability.

Author reply: We would like to highly appreciate the reviewer for this suggestion, and conduct a thorough review of the narrative consistency throughout the manuscript to further enhance the readability. The updated contents are pasted below for viewing convenience:

(1) In the introduction section:

“Due to the energy difference between molecular hybrid orbital energy level of the gaseous molecular and affinity of the sensing block, the p/n-type doping will occur. However, the accompanying chemical reaction on the block surface induces a long-time and complex sensing response²³.”

(2) In the introduction section of the manuscript:

“Importantly, these three sensing mechanisms are based on the surface doping, and the resistive response is outputted.”

(3) In the section of “**Fabrication of Schottky-Junction-based Gas Sensor**” of the manuscript:

“As the temperature increases from 280 K to 320 K, the sample resistance increases. It is indicated that the highly-doped characteristic of the Bi₂O₂Se suppresses the effect of the Schottky barrier height (SBH)²⁸.”

(4) In section “**Molecularly Specific Detection towards Trace Gases**” of the manuscript:

“Regarding the impedance angle, it is dominated by the surface scattering. The absolute value of the impedance angle increases for more adsorbates (*e.g.*, methanol, methanal, and acetone).”

(5) In section “**Molecularly Specific Detection towards Trace Gases**” of the manuscript:

“Towards the 400 ppt NO₂, the response is 3.5/3/6.7 % for the resistive/reactance/impedance signal. Considering that the noise is lower than 0.4 % at the frequency range higher than 0.02 Hz (supplementary information S7), this signal-to-noise ratio indicates that the NO₂ sensing is featured with the detection limit in ppt range. The sensitivity is estimated to 8.8/7.5/16.8 %·ppb⁻¹ for the resistive/reactance/impedance. The NO₂ sensitivity shows orders of magnitude higher than those of the other gases.”

(6) In the section of “**Molecularly Specific Detection towards Trace Gases**” in the manuscript:

“Note that this NO₂ doping manifests as the charge transferring from the Bi₂O₂Se to the NO₂ adsorbed. The NO₂ is therefore served as the adsorbate, rather than the dopant for inducing the reconstruction of the Bi₂O₂Se (supplementary information S8).”

(7) In the section of “**Molecularly Specific Detection towards Trace Gases**” in the manuscript:

“To further study on the molecular specificity, the concentration of the mixed (*e.g.*, methanol or methanal) is increased up to 33% and 50 %.”

Reviewer #2 (Remarks to the Author):

The authors have addressed my concerns towards the previous manuscript. This revised manuscript is acceptable for publication on *Nature Communications*.

Author reply: We would like to highly appreciate the reviewer for this comment.

Reviewer #3 (Remarks to the Author):

The authors carefully revised the manuscript, making all necessary changes based on the reviewers' comments and suggestions.

Author reply: Many thanks to the reviewer for this comment.

Reviewers' Comments:

Reviewer #1:

Remarks to the Author:

The authors prepared the revised manuscript by considering comments raised by reviewers. So, this manuscript can be published as current form in Nature Communications.

A point-by-point response to the reviewer' comments

REVIEWERS' COMMENTS

Reviewer #1 (Remarks to the Author):

The authors prepared the revised manuscript by considering comments raised by reviewers. So, this manuscript can be published as current form in *Nature Communications*.

Author reply: We would like to highly appreciate the reviewer for this positive comment.